**COMMUNICATIONS**

# Evidence for a task-dependent switch in subthalamo-nigral basal ganglia signaling

Jay J. Jantz [1], Masayuki Watanabe[1], Ron Levy[1,2] & Douglas P. Munoz[1,3,4,5]

Basal ganglia (BG) can either facilitate or inhibit movement through excitatory and inhibitory pathways; however whether these opposing signals are dynamically regulated during healthy behavior is not known. Here, we present compelling neurophysiological evidence from three complimentary experiments in non-human primates, indicating task-specific changes in tonic BG pathway weightings during saccade behavior with different cognitive demands. First, simultaneous local field potential recording in the subthalamic nucleus (STN; BG input) and substantia nigra pars reticulata (SNr; BG output) reveals task-dependent shifts in subthalamo-nigral signals. Second, unilateral electrical stimulation of the STN, SNr, and caudate nucleus results in strikingly different saccade directionality and latency biases across the BG. Third, a simple artificial neural network representing canonical BG signaling pathways suggests that pathway weightings can be altered by cortico-BG input activation. Overall, inhibitory pathways (striato-pallidal-subthalamo-nigral) dominate during goal-driven behavior with instructed rewards, while facilitatory pathways (striato-nigral and subthalamo-pallidal-nigral) dominate during unconstrained (free reward) conditions.

[1] Centre for Neuroscience Studies, Queen's University, Kingston, ON, Canada K7L 3N6. [2] Division of Neurosurgery, Department of Surgery, Queen's University, Kingston, ON, Canada K7L 3N6. [3] Department of Biomedical and Molecular Sciences, Queen's University, Kingston, ON, Canada K7L 3N6. [4] Department of Psychology, Queen's University, Kingston, ON, Canada K7L 3N6. [5] Department of Medicine, Queen's University, Kingston, ON, Canada K7L 3N6. Correspondence and requests for materials should be addressed to J.J.J. (email: jay.jantz@queensu.ca)

The basal ganglia (BG) are a group of interconnected sub-cortical nuclei that can influence a variety of functions including motor control, and saccadic eye movements[1] via connections with the cerebral cortex and the superior colliculus (SC; critical for saccade control). Multiple parallel signaling pathways through the BG can either activate or inhibit BG output structures[2]. The striatum and the subthalamic nucleus (STN) receive input signals to the BG; each can inhibit or activate downstream BG output nuclei via different signaling pathways (striatum: 'direct' and 'indirect', respectively[2, 3]; STN: subthalamo-pallidal and 'hyperdirect', respectively[4–6]); and, each have been related to switching between tasks during flexible behavior[7, 8]. Phasic signals through the direct and indirect pathways from the striatum are temporally and spatially separated due to differences in transduction speed and output fiber connectivity, and can be modulated by dopamine in the BG circuit[9–11]. However to the best of our knowledge, it has not been determined whether a similar modulation occurs between hyperdirect and subthalamo-pallidal pathways (Fig. 1a) during behavior[12]. Furthermore, relative tonic strength or weighting between healthy BG pathways is often assumed to be constant across different behavioral conditions, while tonically unbalanced activity between pathways has been associated with diseases such as Parkinson's[13, 14]. We hypothesize that these disorders may describe extremities of a spectrum, in which alterations of the tonic weighting of BG pathways (e.g., inhibitory pathways outweigh facilitatory pathways, or the opposite) may refine healthy voluntary movement control in flexible behavior. Here, we aimed to determine whether the tonic weighting between inhibitory and excitatory pathways from the STN (BG input and relay structure) to the substantia nigra pars reticulata (SNr; BG output structure) may alter according to behavioral context (Fig. 1c). Furthermore, we include and contrast these with published caudate nucleus data in the same monkeys and tasks, to fully encompass BG input and output stages in oculomotor control, and test for coordinated changes across the entire BG network[15].

Examining functional changes across the BG network necessitates a careful and elaborate methodology, because the small size of some BG nuclei limits their temporal and anatomical resolution by functional magnetic resonance imaging (MRI), necessitating more targeted neurophysiological techniques. Many of these neurophysiological techniques are, in turn, too focused to concurrently examine widespread changes across the BG circuit. We have approached this challenge using three complimentary experiments to test network BG activity, during tasks that juxtapose reward-driven and unconstrained free viewing (free reward) behavior requiring very different cognitive demands. First, we examine the phase difference of local field potential signals recorded simultaneously in the STN and the SNr of healthy monkeys. Second, we causally probe the influence of the STN and SNr, with low-current electrical stimulation of each structure in the saccade preparatory period, while using the resultant saccade biases as a behavioral indicator of BG output to the SC. Third, we model BG signaling phase differences using an artificial neural networks model representing the STN, CD, GPe, and SNr. We present evidence for robust differences in BG signaling at the level of the STN: subthalamo-nigral inhibitory output was decreased during the free viewing task, but increased during the rewarded goal-driven task. The inclusion and comparison to caudate nucleus stimulation reveals simultaneous changes within BG direct and indirect pathways. We suggest that during an unconstrained task condition such as free viewing, the BG release tonic inhibitory control from downstream motor structures such as the SC via the disynaptic subthalamo-pallidal-nigral pathway (Fig. 1b, blue pathway) and the 'direct' striato-nigral pathway, consistent with fast automatic or sensory-driven

movements toward unexpected stimuli. In sharp contrast, during a rewarded goal-driven condition, BG increase inhibitory signals via the monosynaptic subthalamo-nigral pathway (Fig. 1b, red pathway) and the 'indirect' striato-nigral pathway, consistent with a reduction of erroneous movements in favor of voluntary motor commands with strong preparatory activity to elicit a saccade accurately[16, 17].

## Results

**Training monkeys on saccade tasks.** We trained two monkeys to perform two very different saccade tasks: an unconstrained free viewing task, and a goal-directed task. The unconstrained task was absent of any local visual stimuli or behavioral cues, and consisted of free viewing on a blank gray screen to capture saccadic eye movements that had no explicit goal (Fig. 1c, upper panel). The goal-directed task consisted of randomly interleaved conditions that required the monkey to either look toward (pro-saccade) or 180° away to the blank screen (anti-saccade) from a peripheral visual cue[8, 18] (Fig. 1c, lower panel). The inclusion of two different instruction conditions in the goal-driven task necessitated the monkey to maintain active engagement in the task. Here, we limit our discussion to the goal-driven anti-saccade condition, as these were the best suited for comparison to free viewing saccades because no visual stimuli were present at the fixation point or saccade target in either task before saccade onset. This allowed a comparison between conditions with different cognitive demands (unconstrained viewing versus explicit task instructions with a goal) while avoiding the confound of visually driven versus internally driven saccades. During these tasks, we performed two experiments to probe functional changes in subthalamo-nigral signaling: first, the simultaneous recording of STN and SNr local field potential (LFP) signals in the saccade preparatory period (Fig. 1a; Experiment 1), and second, low-current electrical stimulation of the STN and SNr in the same period (Fig. 1b; Experiment 2). The time sequence of task events is described in Fig. 1c. In reporting results below, 'ipsiversive' and 'contraversive' refers to behavior (i.e., saccades elicited toward either the same or opposite visual hemifield relative to the recording or stimulation site), whereas 'ipsilateral' and 'contralateral' refers to anatomy (i.e., brain regions located in either the same or opposite brain hemisphere relative to the recording or stimulation site).

**STN-SNr LFP phase angle changes between tasks.** We simultaneously recorded STN and SNr LFP activity using acute electrode pairs, while two monkeys performed free viewing and goal-directed tasks (Fig. 1a; Experiment 1). We examined the 200 msec pre-saccadic period in both tasks, during which no visual stimuli were present at the fixation point or saccade target. We tested whether there were task-dependent changes in subthalamo-nigral signaling by comparing within-site differences in the coherence and phase angle between STN and SNr LFP signals. For example, when comparing two signals $x(t)$ and $y(t)$ (e.g., Fig. 1a, pathways 1 and 2, respectively), if $x = y$ then the coherence between signals $x$ and $y$ equals 1 with a phase angle of 0. However, if $x = -y$, the coherence between $x$ and $y$ would still equal 1, but the phase angle would change to 180°. If there are task-dependent alterations in the tonic weighting of BG pathways (such as between the hyperdirect and subthalamo-pallidal-nigral pathways), we hypothesized that this would result in a phase angle change based on different signal transduction times, and the 180° signal phase shift in signals projected through the pallido-nigral GABAergic (gamma-aminobutyric acid) pathway (Fig. 1a, top panel). Importantly however, we would expect to observe a sub-180° shift in phase angle overall, because of competing signals between the

hyperdirect and subthalamo-pallidal-nigral pathways, as well as simultaneous striato-nigral effects on the SNr. Therefore, we predict that a dominance of the hyperdirect pathway should be associated with a positive STN-SNr phase difference (based on glutamatergic subthalamo-nigral projections and a short signal transduction time), while a dominance of the subthalamo-pallidal-nigral pathway should be associated with a negative STN-SNr phase difference (when measured peak-to-peak; based on a 180° signal phase shift for subthalamo-pallidal-nigral signals, and an increased transduction time; Fig. 1a, bottom panel).

There was strong coherence between STN and SNr signals encompassing the beta frequency band during both the free viewing and the goal-directed anti-saccade task; from 5 to 29 Hz during anti-saccades and from 5 to 48 Hz during free viewing saccades. However, STN-SNr coherence was significantly higher in the free viewing task than the anti-saccade task from 31 to 58 Hz (Wilcoxon Rank-Sum Test, $p < 0.05$, $n = 15$; Fig. 2a). Coherence line widths in Fig. 2a reflect population standard error around the mean recorded at each frequency. Dashed horizontal lines indicate the 99% confidence line for an individual site ($p < 0.05$), and asterisks indicate the STN-SNr phase angle at each frequency at which coherence was significant. Individual site coherence is shown in Supplementary Fig. 2. Figure 2b demonstrates within-site differences in phase angle between free

viewing and anti-saccades in the 15–25 Hz peak coherence window. Strikingly, there was a negative phase difference between STN and SNr signals during free viewing saccades, but a positive phase difference during anti-saccades (Wilcoxon Rank-Sum Test, $n = 15$ sites, 2523 free viewing saccade trials (after stringent filtration of saccade eccentricity and fixation duration), 4882 correct anti-saccade trials, $p < 0.05$). This suggests that the weighting between subthalamo-nigral pathways may have indeed changed between tasks. When movement was unconstrained during free viewing, the subthalamo-pallidal-nigral pathway (e.g., Fig. 1a, pathway 1) may have dominated, while when movement was goal driven during anti-saccades, the hyperdirect pathway may have dominated (e.g., Fig. 1a, pathway 2). However, task-dependent phase angle differences were <180°, implicating a weighting shift (not an absolute switch) potentially between the hyperdirect and subthalamo-pallidal-nigral pathways, as well as likely involvement of indirect pathway signals. This task-dependent change in phase angle is further modeled using a spiking neural networks model at the conclusion of the paper. However first, to causally test whether there was a task-dependent change in effect of subthalamo-nigral signaling, we compared the consequence of SNr and STN electrical stimulation between free viewing and anti-saccade initiation.

**SNr stimulation inhibited contraversive or bilateral saccades**. SNr stimulation inhibited contraversive or bilateral free viewing saccades (Fig. 1b; Experiment 2). Eye movements were unrestricted during the free viewing task and therefore 'trials' began when the monkey happened to fixate near the center of the

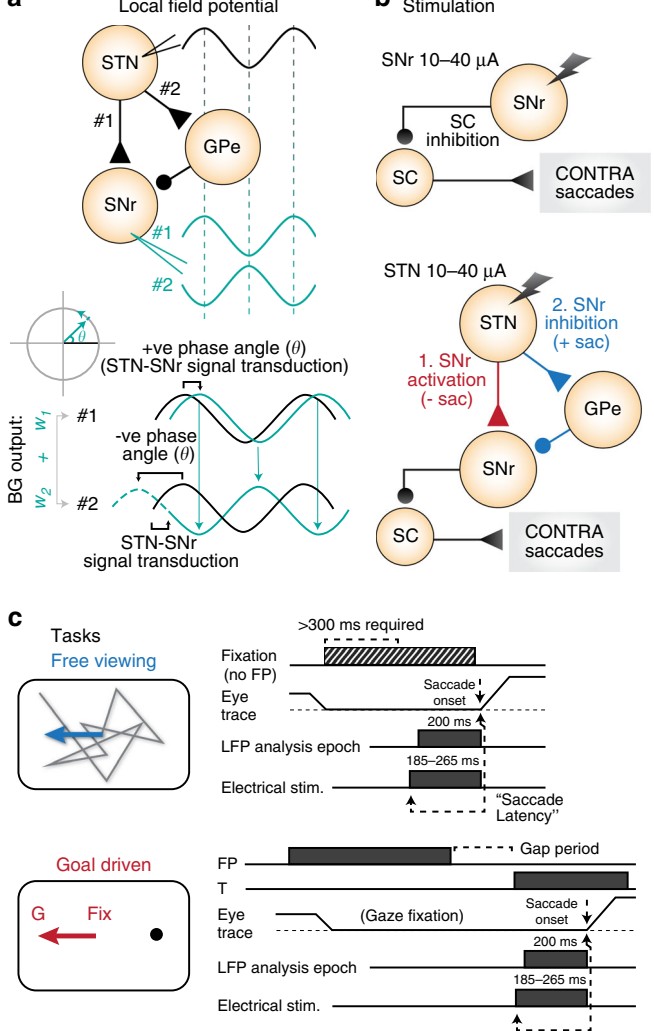

**Fig. 1** Subthalamo-nigral unilateral connectivity, saccade tasks, and experimental outline. **a** Experiment 1: STN and SNr local field potential was recorded simultaneously in both structures, $n = 15$ acute electrode pairs. The subthalamo-nigral "hyperdirect" pathway should produce a small positive phase angle between STN and SNr signals due to signal transduction delay, while the subthalamo-pallidal-nigral pathway should produce a negative phase angle (measured from peak to peak) because of a 180° inversion of signals from GABAergic GPe-SNr projections. **b** Experiment 2: STN and SNr electrical stimulation. The SNr (BG output) sends GABAergic inhibitory fibers directly to the ipsilateral SC (to inhibit contraversive saccades), as well as to a lesser extent the contralateral SC[8, 20, 31, 32] (not shown). The SC influences contraversive saccades via the saccade generating circuit in the brainstem[33]. STN electrical stimulation should have a similar effect as SNr stimulation if the subthalamo-nigral (red) pathway predominates (i.e., STN activates the SNr), while STN stimulation should have different or opposite effects as SNr stimulation if the subthalamo-pallidal-nigral (blue) pathway predominates (i.e., STN inhibits the SNr)[4, 5, 34]. Circular and triangular endpoints reflect inhibitory and excitatory projections, respectively. The STN also relays striato-pallidal "indirect" pathway signals to BG output nuclei (not shown here; Figs. 6 and 7). GPe, external segment of the globus pallidus; SC, superior colliculus; SNr, substantia nigra pars reticulata; STN, subthalamic nucleus. **c** Two monkeys performed an unconstrained free-viewing task and a goal-driven anti-saccade task (saccade 180° away from visual target)[15, 18]. No visual stimuli existed at the fixation point or saccade target before saccade onset in either task. Free-viewing was entirely absent of visual cues or stimuli. Free-viewing "trials" were defined by the monkey fixating (by chance) for at least 300 ms, ±10° from the center of the screen (Methods section). Anti-saccades were also interleaved with pro-saccade (look toward) trials to maintain active engagement in the task (Supplementary Figures). Task events, eye movements, pre-saccadic LFP recording period (200 ms), and pre-saccadic stimulation periods are illustrated. Saccade latency was defined in all tasks as the time from electrical stimulation onset to saccade onset. FP, visual fixation point; G, saccade goal; T, visual target

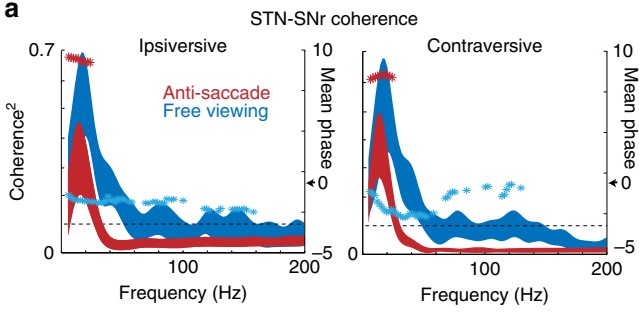

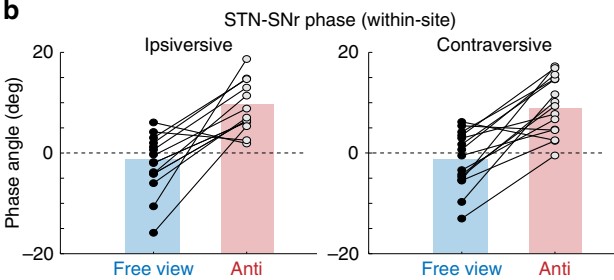

**Fig. 2** STN-SNr coherence and phase angle during the 200 msec period before free viewing saccade (blue) and anti-saccade (red) onset. No visual stimuli were present at the fixation point or saccade target during this period in either task (i.e., unconstrained and free reward during free viewing; goal directed and reward driven during anti-saccades. **a** STN-SNr coherence across frequencies, line width reflects population standard error, and horizontal dashed lines reflect the 99% confidence line against 0. STN and SNr LFP was coherent from 5 to 29 Hz before anti-saccade onset, and from 5 to 48 Hz before free viewing saccade onset. STN-SNr coherence was significantly higher during the free viewing saccade task than anti-saccade task, from 31 to 58 Hz (Wilcoxon Rank-Sum Test, $p < 0.05$). Asterisks indicate STN-SNr LFP phase angle at frequencies when STN-SNr coherence is above 99% confidence. **b** STN-SNr LFP phase angle is compared within sites, in the 15–25 Hz frequency window encompassing peak coherence. There was a negative phase angle between STN and SNr signals during free viewing, but a positive phase angle during anti-saccades, consistent with a weighting shift in subthalamo-pallidal pathways

screen. During control trials without stimulation, monkeys made randomly distributed saccades across the screen, with mean horizontal and vertical saccade endpoints distributed around center (e.g., Fig. 3a, gray points). Low-current (unilateral) SNr stimulation was applied during the pre-saccadic period, with timing comparable to the LFP analysis period described above (see Fig. 1c, and Methods section). This began 300 ms after the monkey fixated near the center of the screen, and ended when a saccade was detected or after 800 ms. SNr stimulation either inhibited saccades directed toward the contraversive visual hemifield (Fig. 3b, blue arrows), or inhibited saccades bilaterally (Fig. 3b upper left panel, teal arrow and trace). Proportions of all SNr and STN stimulation site effects are summarized in Supplementary Table 1. Contraversive or bilateral saccade inhibition was defined at each site based on the change in ipsiversive or contraversive saccade latency. Stimulation sites that inhibited saccades bilaterally ($n = 21/68$) were analyzed separately, because they exhibited little or no saccade direction bias toward ipsiversive or contraversive hemifields, and skewed the mean cumulative distribution of saccade latencies when grouped with other sites. We first address all other stimulation sites ($n = 47/68$). As described below, this population resulted in an overall contraversive inhibition of saccades. Figure 3a (upper left panel) illustrates the effect of SNr stimulation at one site at which saccade

endpoint vectors were biased toward the ipsiversive (left) visual hemifield. We calculated the difference in mean saccade direction between control and stimulation trials at each site (Methods section), to compare the effect of stimulation across the population in 2 monkeys (Supplementary Fig. 1 for anatomical localization). Stimulation across this population biased saccade endpoints ipsiversively overall (Fig. 3a, upper right panel; monkey E: $t_{(7)} = -5.62$, $p < 0.0001$; monkey O: $t_{(37)} = -5.87$, $p < 0.0001$ (paired $t$-test)), and increased contraversive saccade latency (Fig. 3b; paired $t$-test, $t_{(46)} = -2.35$, $p < 0.05$), consistent with inhibition of the downstream ipsilateral SC[19, 20]. Moreover, SNr stimulation increased ipsiversive saccade frequency (Supplementary Fig. 3a; Two-sample Kolmogorov–Smirnov test, $D = 0.51$, $p < 0.001$, $n = 47$), and decreased contraversive saccade frequency (Supplementary Fig. 3b; Two-sample Kolmogorov–Smirnov test, $D = 0.63$, $p < 0.001$, $n = 47$) across the range of ipsiversive and contraversive free viewing saccade latencies, ruling out a trend driven by only short- or long-latency saccades. Some SNr stimulation sites also vertically biased saccades upward, but this was significant only in Monkey O (monkey E: $t_{(7)} = 1.38$, $p = 0.22$; monkey O: $t_{(37)} = 9.22$, $p < 0.001$ (paired $t$-test); $n = 10$).

SNr stimulation sites that inhibited free viewing saccades bilaterally ($n = 21/68$) were analyzed separately. We calculated the cumulative distribution of saccade frequency at each of these stimulation sites, and then averaged across all sites (Supplementary Fig. 3c, d). The mean cumulative distribution of saccade latencies confirmed a bilateral decrease in saccade frequency (ipsiversive: $D = 0.50$, $p < 0.001$, $n = 21$; contraversive: $D = 0.55$, $p < 0.001$, $n = 21$ (Two-sample Kolmogorov–Smirnov test)). Stimulation at these sites also increased ipsiversive saccade latency (Fig. 3b, teal line, paired $t$-test, $t_{(20)} = -2.11$, $p < 0.05$). These sites are consistent with activation of both uncrossed and crossed GABAergic SNr efferents[20], inhibiting the SC bilaterally as reported previously[16]. There were no systematic differences in the coordinates of stimulation sites between contraversive and bilateral suppression. Of the SNr stimulation sites that significantly affected saccade initiation ($n = 45/68$), 40% ($n = 18$) inhibited contraversive saccades, and 47% ($n = 21$) inhibited saccades bilaterally.

In the goal-driven anti-saccade task, SNr stimulation increased latencies bilaterally (ipsiversive: $t_{(77)} = 3.85$, $p < 0.001$; contraversive: $t_{(77)} = 7.91$, $p < 0.001$ (paired $t$-test); Fig. 4a), which occurred in 64% of effective stimulation sites (Fig. 4a bottom left quadrant; $p < 0.05$, paired $t$-test, $n = 39$). However, consistent with SNr stimulation during the free viewing task, anti-saccade contraversive latencies remained significantly higher than ipsiversive latencies (Fig. 4a, b; paired $t$-test, $t_{(154)} = 2.97$, $p = 0.004$), suggesting activation of uncrossed, and to a lesser extent crossed, GABAergic nigrotectal projections[20]. Specifically, 88% of SNr stimulation sites with a significant effect were also associated with significantly greater contraversive than ipsiversive inhibition of saccades (Fig. 4a above the line of unity; paired $t$-test, $p < 0.05$, $n = 39$). Additionally, both cumulative ipsiversive saccade frequency (Supplementary Fig. 4a; anti-saccade: $D = 0.21$, $p < 0.001$ (Two-sample Kolmogorov–Smirnov test), $n = 39$), and contraversive saccade frequency (Supplementary Fig. 4b; anti-saccade: $D = 0.28$, $p < 0.001$ (Two-sample Kolmogorov–Smirnov test), $n = 39$), was shifted right across the range of saccade latencies. All SNr stimulation sites with bilateral inhibition in the free viewing task also exhibited bilateral inhibition in the goal-driven task, consistent with activation of both crossed and uncrossed GABAergic nigrotectal projections to inhibit the downstream SC.

**Effect of STN stimulation was task dependent.** STN stimulation facilitated contraversive saccades during free viewing. In Fig. 3a

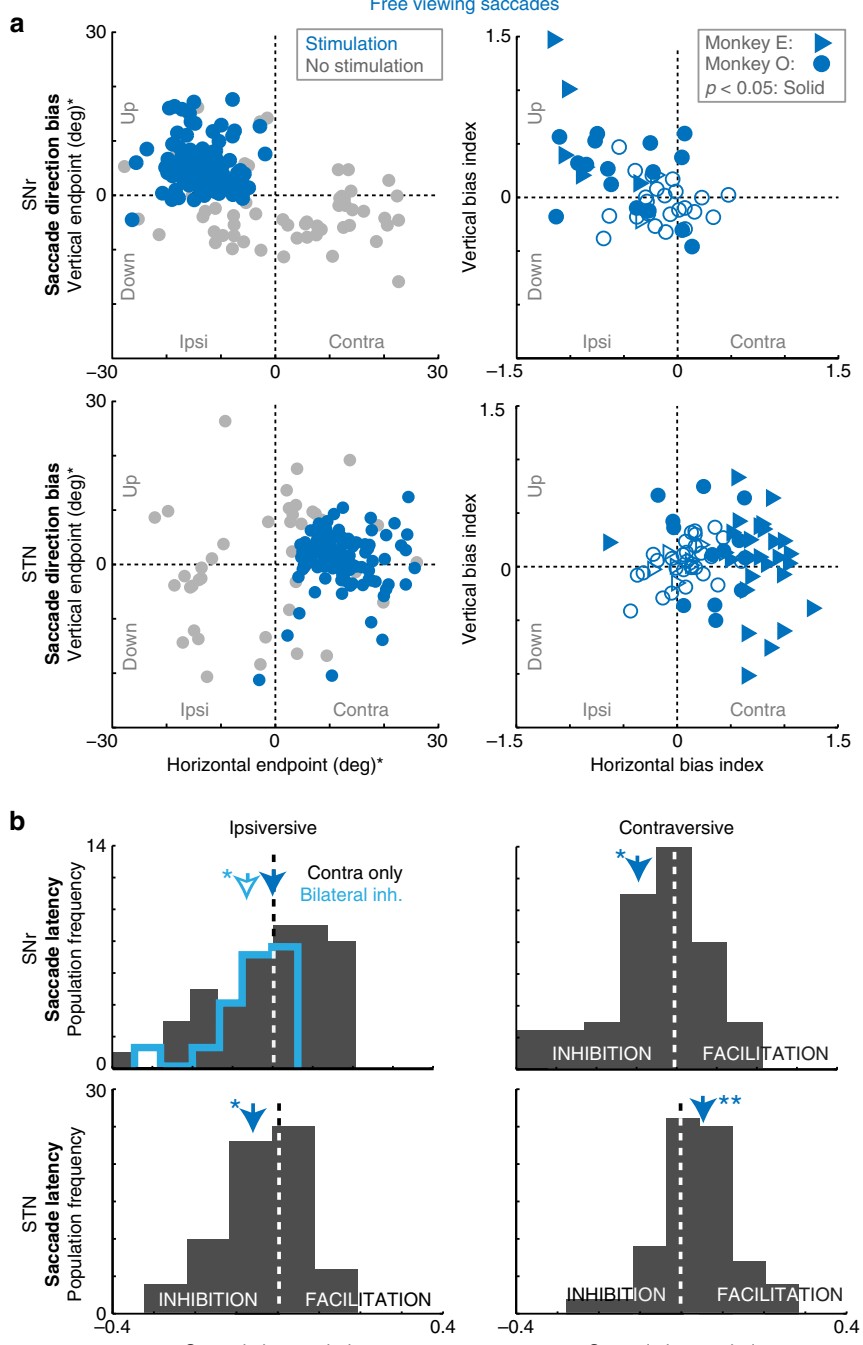

**Fig. 3** SNr stimulation contraversively inhibited or bilaterally inhibited free viewing saccades, while STN stimulation contraversively facilitated free viewing saccades. Electrical stimulation was delivered across the left STN or SNr of 2 monkeys. In the SNr, 66% of stimulation sites affected saccade direction or frequency (paired t-test, $p < 0.05$). Of these, 40% were associated with an ipsiversive (left) bias of free viewing saccades with respect to the stimulated left hemisphere, whereas 47% were associated with a bilateral decrease in saccade frequency. In the STN, 62% of stimulation sites affected saccade direction or frequency (paired t-test, $p < 0.05$). Of these, 74% were associated with a contraversive bias of saccades, where saccade biases (i.e., contraversive facilitation, bilateral inhibition, etc.) were defined based on a change in saccade frequency toward ipsiversive or contraversive visual hemifields. **a** Left panels: representative SNr and STN stimulation sites, indicating an ipsiversive and contraversive bias of saccades, respectively. Each point indicates the end point of a single saccade (saccade start positions normalized to the origin), with overlaid control (gray) and stimulation (blue) trials. Saccade start points were standardized to the origin. Right panels: population direction bias index. Average saccade bias was quantified for each stimulation site, each point represents a single site. Filled points are significantly different from 0 (paired t-test, $p < 0.05$). Paired t-test, SNr: $p < 0.001$ both monkeys, $n = 47$; STN: $p < 0.001$ both monkeys, $n = 76$. **b** Population histograms of change in ipsiversive (left panels) and contraversive (right panels) saccade latency after stimulation. Negative and positive values indicate increased and decreased saccade latency after stimulation, respectively. *significant difference (paired t-test $p < 0.05$). **significant difference (paired t-test $p < 0.001$). Teal arrows and trace indicate the subset of SNr stimulation sites that bilaterally inhibited spontaneous saccades ($n = 21$). These were analyzed separately, because there was no appreciable saccade direction bias toward either visual hemifield

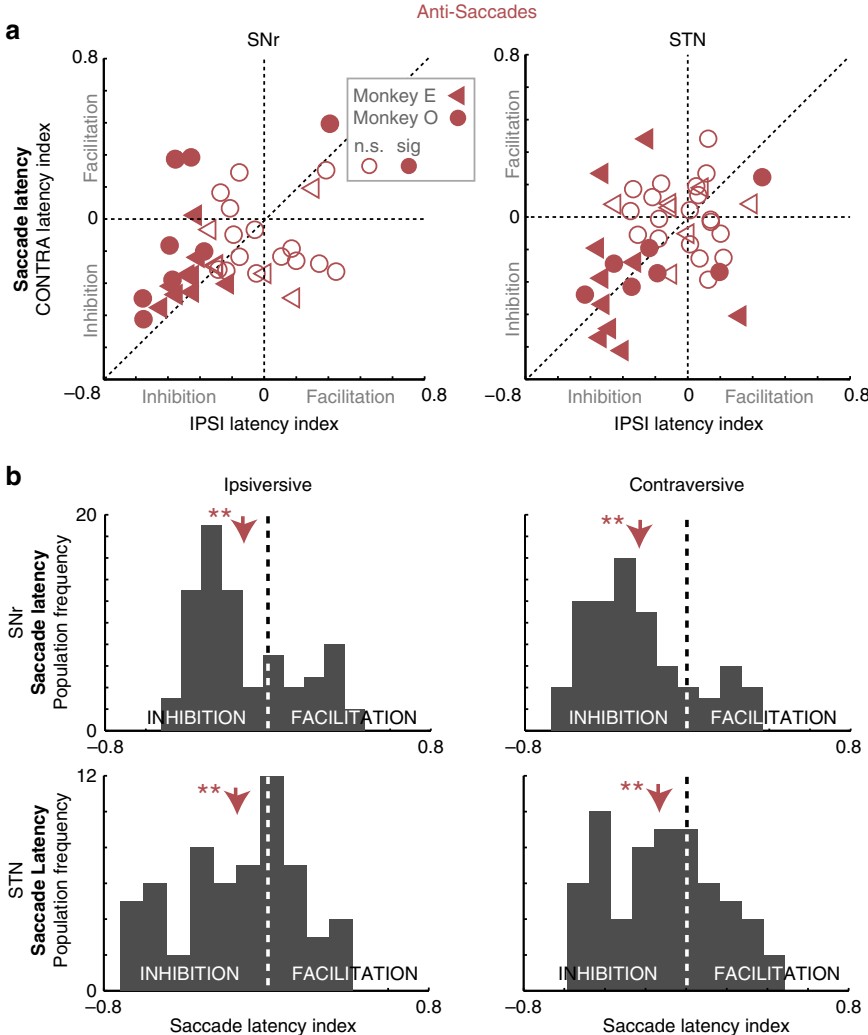

**Fig. 4** SNr and STN stimulation both inhibited anti-saccades bilaterally. **a** Population saccade latency index after SNr or STN stimulation. Each point represents a single stimulation site (120 trials minimum). Filled points are significantly different from 0 (paired $t$-test, $p < 0.05$). n.s. not significant, sig significant. **b** Population histograms of change in ipsiversive (left panels) and contraversive (right panels) saccade latency after SNr or STN stimulation. Negative and positive values indicate increased and decreased saccade latency after stimulation, respectively. **significant difference (paired $t$-test $p < 0.001$)

the bottom left panel illustrates the effects of STN stimulation at one site and the bottom right panel compares the ipsiversive and contraversive saccade biases for 76 STN stimulation sites in 2 monkeys (see Supplementary Fig. 1 for anatomical localization). Overall, STN stimulation biased free viewing saccades contraversively [monkey E: $t_{(32)} = -4.47$, $p < 0.0001$; monkey O: $t_{(40)} = -7.33$, $p < 0.001$ (paired $t$-test)], which was the opposite of SNr stimulation. Specifically, STN stimulation increased ipsiversive saccade latency (Fig. 3b, bottom left panel; inhibition; paired $t$-test, $t_{(75)} = 2.86$, $p = 0.005$), and decreased contraversive saccade latency (Fig. 3b, bottom right panel; facilitation; paired $t$-test, $t_{(75)} = -2.30$, $p = 0.02$), suggesting an inhibition of ipsiversive saccades and facilitation of contraversive saccades. The latency of spontaneously generated saccades during free viewing was defined as the period of fixation preceding the first saccade initiated during arbitrarily defined 'trials' (beginning 300 ms after eyes entered the central window; i.e., time-locked to electrical stimulation onset; see Fig. 1c). Saccade frequency was decreased in the ipsiversive direction (Supplementary Fig. 3e; Two-sample Kolmogorov–Smirnov test, $D = 0.51$, $p < 0.001$, $n = 76$), and increased in the contraversive direction (Supplementary Fig. 3f;

Two-sample Kolmogorov–Smirnov test, $D = 0.54$, $p < 0.001$, $n = 76$) across the range of free viewing saccade latencies (i.e., effects were not driven by only short- or long-latency saccades).

STN stimulation revealed strikingly different effects on anti-saccades, when monkeys were required to follow specific instructions to achieve a rewarded goal (Methods section), than free viewing saccades. Here, stimulation during saccade preparation inhibited saccades bilaterally (Fig. 4b, lower panels; increased ipsiversive saccade latencies: $t_{(85)} = 4.04$, $p < 0.001$; increased contraversive saccade latencies: $t_{(88)} = 4.22$, $p < 0.001$ (paired $t$-test)), with no significant difference in latencies between hemifields. Additionally, STN stimulation decreased saccade frequency bilaterally (Supplementary Fig. 4c, d; ipsiversive: $D = 0.48$, $p < 0.001$; contraversive: $D = 0.31$, $p < 0.001$ (Two-sample Kolmogorov–Smirnov test), $n = 45$) across all anti-saccade latencies (i.e., effects were not driven by only short- or long-latency saccades). Therefore, STN stimulation had explicit task-specific effects between unconstrained free viewing and goal-directed anti-saccade conditions, which importantly, occurred within the same STN stimulation sites ($n = 41$). During free viewing, STN stimulation produced a decrease in contraversive

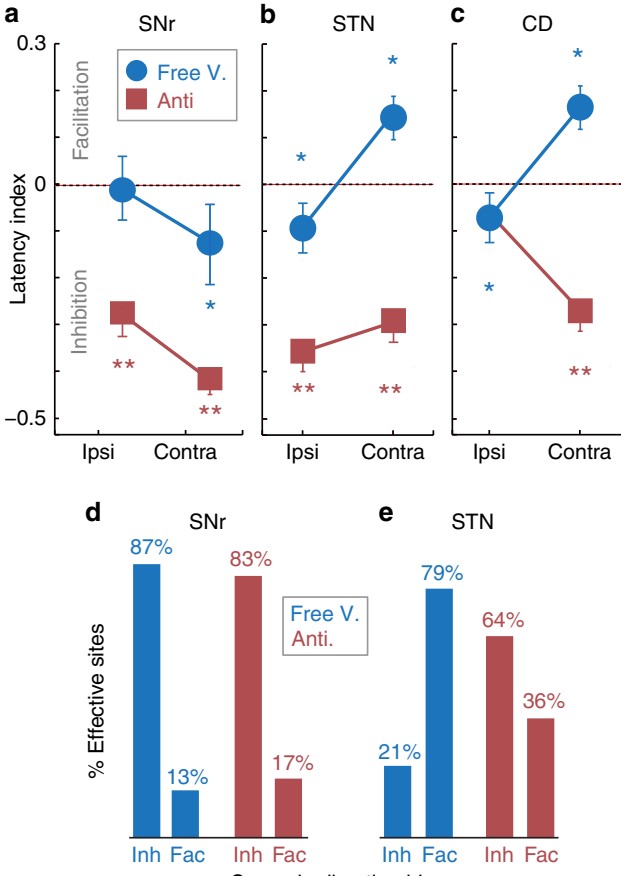

**Fig. 5** Summary of effects. **a–c** SNr, STN, and caudate nucleus stimulation effects on saccade latency during free viewing (blue) and anti-saccade (red) conditions. Caudate nucleus and STN stimulation resulted in task-specific saccade direction biases while the SNr did not; caudate and STN direction biases were opposite to SNr direction biases during free viewing saccades (i.e., contraversive facilitation versus inhibition), but comparable to the SNr during goal-driven anti-saccades (i.e., bilateral inhibition). Error bars indicate standard error. **d, e** A summary of % sites with a significant difference in saccade frequency after stimulation (paired *t*-test, $p < 0.05$) supports task-specific effects of the STN. Inhibition (inh) describes any site with either an ipsiversive saccade bias (defined by an increased frequency of ipsiversive saccades, or a greater decrease in contraversive than ipsiversive saccade frequency), or a bilateral decrease in saccade frequency. Facilitation (fac) describes any site with either a contraversive saccade bias (defined by an increased frequency of contraversive saccades, or a greater decrease in ipsiversive than contraversive saccade frequency), or a bilateral increase in saccade frequency

saccade latency, and biased saccades toward the contraversive hemifield. In sharp contrast, STN stimulation during goal-directed anti-saccades increased latencies bilaterally.

**Summary of results**. Figure 5 compares the saccade latency effects of STN to SNr stimulation during free viewing and goal-directed anti-saccade conditions, and includes caudate nucleus stimulation effects with the same tasks, stimulation parameters, and monkeys that were published previously in another format[8, 15]. Caudate (CD) nucleus and STN stimulation both biased free viewing saccades contraversively, while stimulation at the same sites inhibited or delayed contraversive anti-saccades in the caudate nucleus (Fig. 5c) and bilateral anti-saccades in the

STN (Fig. 5b). Conversely, in both free viewing and goal-directed anti-saccade tasks, SNr stimulation inhibited saccades bilaterally, and/or biased saccades away from the contraversive visual hemifield (Fig. 5a). A task-specific effect at the level of the STN is also apparent when considering the respective proportions of stimulation sites that significantly affected saccade frequency in each structure (Fig. 5d, e). By classifying stimulation effects broadly into 'Inh' (bilateral or contraversive inhibition) and 'Fac' (bilateral or contraversive facilitation) categories to generalize the STN and SNr results described in detail above, the opposite effects of STN and SNr stimulation during free viewing (Fig. 5d, e, blue bars) but comparable effects during goal-driven anti-saccades (Fig. 5d, e, red bars) are clearly observable, supporting explicit task-dependent changes within the BG.

Here, we limited our discussion to the goal-directed anti-saccade condition as this was the best suited for comparison to free viewing saccades (because of the absence of a visual stimulus at both the fixation point and saccade target, in both tasks). However, the interested reader can examine the pro-saccade results in Supplementary Figs. 2, 4, and 5. The pro-saccade results were qualitatively similar to the anti-saccade results, but with a smaller absolute magnitude of effect. Importantly, while anti-saccades require the withholding of an automatic visually guided saccade and pro-saccades do not, both require saccades to be directed toward a specific target to receive a reward. This is in sharp contrast to free viewing behavior, which was voluntarily executed, but not associated with any explicit reward. This suggests that the shift in BG pathway weightings observed here might relate to unconstrained versus goal-directed saccade behaviors, rather than automatic versus voluntary behavior.

Altogether during free viewing, the negative STN-SNr LFP phase angle from Experiment 1 (Figs. 1a, 2 blue traces) and the opposite effects of stimulation of the STN and SNr from Experiment 2 (Figs. 1b and 3) are consistent with a higher weighting of the subthalamo-pallidal-nigral pathway (e.g., STN inhibits the SNr; Fig. 1, blue pathway). In sharp contrast, during goal-driven anti-saccades, the positive STN-SNr LFP phase angle from Experiment 1 (Figs. 1a, 2 red traces) and the comparable effects of STN and SNr stimulation from Experiment 2 (Figs. 1b and 4) are consistent with a higher weighting of the hyperdirect subthalamo-nigral pathway (e.g., STN activates the SNr; Fig. 1, red pathway).

## Discussion
The free viewing and goal-directed anti-saccade tasks represented two extremes of cognitive demand. In both tasks the saccade motor command was internally driven and not guided by a visual stimulus at the saccade goal[15]. However, goal-driven saccades necessitated explicit task instructions to achieve a reward, while free viewing saccades were unconstrained and reward was given freely. Here, we presented converging evidence from two experiments indicating task-specific differences in subthalamo-nigral signaling. First, we simultaneously recorded LFP activity in the STN and SNr, and found that the phase angle of LFP signals between the STN and SNr changed according to task. Second, we electrically stimulated the STN or SNr in the same tasks, while using the resultant saccadic eye movement biases as a proxy measure of BG output activity to the SC. STN stimulation resulted in quantitative and qualitative task-specific saccade biases (i.e., contraversive facilitation versus bilateral inhibition), while SNr stimulation downstream resulted in a consistent saccade direction bias in both tasks (i.e., contraversive and/or bilateral inhibition). Taken together, the incorporation of two independent experiments, and the involvement of three nuclei spanning the input (STN and caudate nucleus) and the output

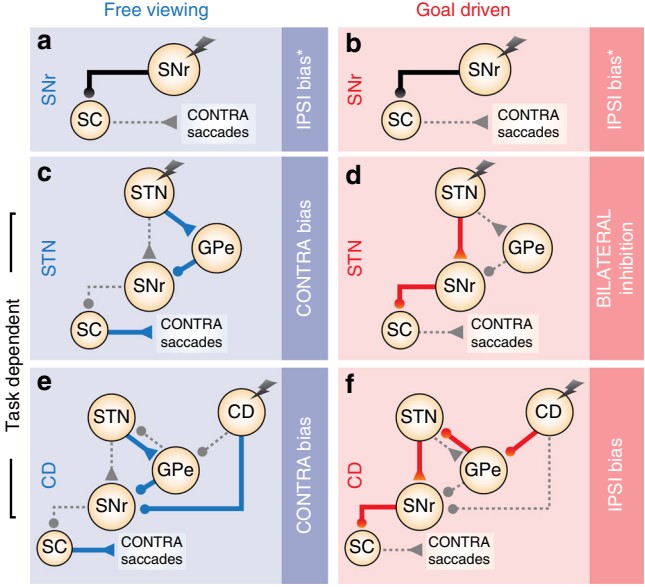

**Fig. 6** Hypothesized state of BG connectivity between spontaneous and anti-saccade conditions. **a**, **b** SNr stimulation either inhibits contraversive saccades (i.e., ipsiversive bias) *or inhibits saccades bilaterally, regardless of free viewing or anti-saccade task condition. **c–f** Both the STN and the caudate nucleus (CD; previously electrically stimulated using the same saccade tasks, stimulation protocols, and monkeys[15] activate the SNr during anti-saccades (red), but inhibit the SNr during free viewing saccades (blue), supporting concerted task-dependent changes across the BG circuit, such that faciliatory subthalamo-pallidal-nigral and striato-nigral "direct" pathways predominate during unconstrained (free reward) free viewing, while inhibitory subthalamo-nigral "hyperdirect" and striato-pallidal-nigral "indirect" pathways predominate during goal-directed anti-saccades. Circular and triangular endpoints reflect inhibitory and excitatory endpoints, respectively

(SNr) stages of the BG in two monkeys, provides compelling evidence for coordinated task-dependent changes in the weighting of BG connectivity between unconstrained and reward-driven behavior.

The STN can anatomically activate opposing pathways to the SNr[4–6] (Fig. 1). We chose simultaneous LFP recording of the STN and SNr because it reflects the summation of multiple local electrical fields[21, 22], and might therefore capture signals from both the subthalamo-nigral and subthalamo-pallidal-nigral pathways in the same pair of electrode recording sites[4–6]. This was necessary to reveal within-site differences in the weighting of subthalamo-nigral pathways that increase (e.g., monosynaptic hyperdirect pathway) or decrease (e.g., disynaptic subthalamo-pallidal-nigral pathway[2]) BG inhibitory output. We calculated STN-SNr coherence across frequencies, and extracted the phase angle between STN and SNr signals at each frequency using the Fourier transform of the cross covariance function[23]. In both tasks, STN and SNr LFP signals were coherent in frequencies encompassing the beta frequency band (13–30 Hz), but revealed distinct within-site differences in phase angle between nuclei according to task condition.

The involvement of beta frequencies in subthalamo-nigral coherence may be reasonable considering evidence that recurrent STN-GPe projections are implicated in developing beta oscillations in the BG[24–27]. However, LFP recording is susceptible to contamination from multiple signals. Furthermore, STN to globus pallidus internus (GPi) coherence exists in the ~20 Hz beta range in human Parkinson's disease patients when off

medication, but this is shifted to ~70 Hz gamma range when on medication[28]. While we did observe significantly increased low-gamma (~31–58 Hz) coherence during unconstrained free viewing saccades versus the presumably more inhibited anti-saccades, we cannot rule out the possibility of STN-SNr beta coherence being driven by artifact in the LFP recording, or by a dopamine reduced state due to overtraining (boredom). However, the specific frequency of coherence largely does not address the hypothesis proposed here, which is instead well supported by the more robust within-site difference of LFP phase angle, and task-dependent effects of STN versus SNr electrical stimulation.

BG stimulation effects provide causal evidence for coordinated task-specific changes, across the STN, SNr, and caudate nucleus. Based on this, likely associated changes in BG functional connectivity are summarized in Fig. 6. Because the saccade control circuit is well-defined, and receives BG output signals from the SNr[10, 19, 29, 30], overall BG output activity during saccade tasks can be described by straightforward behavioral predictions of unilateral electrical microstimulation: either an ipsiversive or contraversive saccade bias, depending on whether the downstream SC is inhibited or disinhibited by SNr GABAergic output (Fig. 1b). In both tasks, we found that SNr stimulation either inhibited saccades bilaterally as previously demonstrated[16], or biased saccades away from the contraversive visual hemifield, consistent with more prominent and/or synaptically stronger ipsilateral nigrotectal GABAergic projection fibers than contralateral nigrotectal GABAergic projection fibers[10, 20, 31, 32] (Fig. 6a, b). Although SNr stimulation saccade bias directions were qualitatively the same between tasks, there were quantitative differences in the magnitude of effect between tasks (Fig. 5a, d). A full explanation is outside the scope of the current study, but may relate to differences in converging input to the downstream SC from structures outside the BG (e.g., frontal eye field, supplementary eye field)[33]. As described above, in contrast to the SNr, STN stimulation revealed dramatically different effects on saccade control between tasks. In the BG oculomotor loop, the two most prominent projections from the STN are glutamatergic efferents to the SNr (i.e., subthalamo-nigral pathway), and to the GPe (i.e., subthalamo-pallidal-nigral pathway), and these have antagonistic effects on saccade initiation via the SC[4, 5, 34]. During free viewing saccades, STN stimulation effects were opposite to SNr stimulation effects (e.g., Fig. 5b, e), suggesting STN-mediated deactivation of the more prominent ipsilateral (uncrossed) nigrotectal projection fibers (Fig. 6c; subthalamo-pallidal-nigral pathway). In sharp contrast, both STN and SNr stimulation inhibited anti-saccades bilaterally (Fig. 6d; subthalamo-nigral pathway), consistent with STN-mediated excitation of the both ipsilateral (uncrossed) and contralateral (crossed) nigrotectal projections from the SNr.

Based on our results, we predict concerted changes across the entire BG network according to behavioral condition. In fact, strong evidence within the same monkeys supports this view. The head and body of the caudate nucleus (BG input) was previously stimulated using the same behavioral conditions, stimulation protocols, and in the same monkeys[15] (Fig. 5c). Like the STN, caudate stimulation effects were task-specific: free viewing saccades were biased contraversively, but goal-driven anti-saccades were biased ipsiversively. The comparison to our SNr stimulation results can now also implicate associated changes in striato-nigral functional connectivity (Fig. 6e, f). In the BG oculomotor loop, the two most prominent projections from the caudate nucleus are GABAergic efferents to the SNr (i.e., striato-nigral "direct" pathway), and to the GPe (i.e., striato-pallidal-subthalamo-nigral "indirect" pathway[2, 3]), and these have antagonistic effects on saccade initiation. The opposite effects of caudate nucleus and SNr stimulation during free viewing implicates a striatal pathway

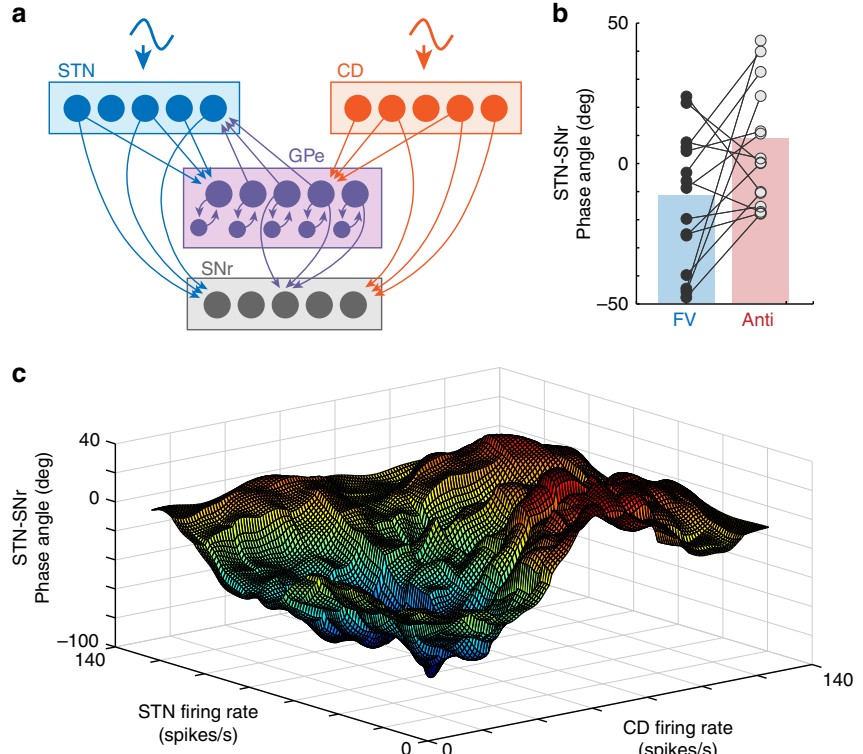

**Fig. 7** Basal ganglia neural networks model of STN-SNr phase angle according to STN and CD activation. A simple spiking neural networks model of the BG was created to test whether altering the input activation of the STN and CD could result in an STN-SNr phase angle shift. Each nucleus ("node") contained 1000 artificial neurons (McCulloch-Pitts neuron[54, 55]), which projected to other nuclei according to major BG projection pathways[1–3, 11, 31]. Anatomical localizations of projection neurons within each nucleus were assigned randomly. **a** Color-coded schematic of BG spiking neural networks model architecture, indicating all modeled projection pathways (hyperdirect, direct, indirect, subthalamo-pallidal-nigral, and recurrent GPe inhibitory layer). **b** Phase angle differences in artificial STN-SNr signals was similar to those observed in LFP recordings (Fig. 2b), at published estimates of STN[7, 56] and CD[8, 57, 58] activities in free viewing and goal-directed saccade conditions (STN = 3 Hz/45 Hz, CD = 3 Hz/20 Hz, for free viewing and goal-directed conditions, respectively). **c** STN-SNr phase angle across all combinations of STN and CD firing rates from 0 to 100 spikes/s revealed a robust shift in STN-SNr phase angle between low and high STN/CD activation levels

that inhibited the SNr (e.g., Fig. 6e; striato-nigral "direct" pathway), while the comparable effects of caudate nucleus and SNr stimulation during goal-directed anti-saccades implicates pathway that excited the SNr downstream (e.g., Fig. 6f; striato-pallidal-subthalamo-nigral "indirect" pathway). Overall, this supports coordinated task-dependent changes in the weighting of pathways across the BG circuit during healthy behavior: both the STN and the caudate nucleus projects to facilitatory and to inhibitory BG pathways[2–5, 34], and a higher relative weighting of the former is implicated during free viewing, and of the latter during anti-saccades. With respect to saccade control, we suggest the conditions in Fig. 6 could prime the saccade control circuit for fast automatic saccades during unconstrained free viewing, and slower accurate voluntary saccades during goal-driven anti-saccades. During goal-directed movement, inhibition of BG output may decrease unnecessary or sub-optimal movements, in favor of correct rewarding movements[16, 17], while during unconstrained free viewing with no explicit goal, reduced tonic inhibition from BG output may facilitate (or disinhibit) automatic movements toward unexpected stimuli. In fact, both the STN and CD have been previously related to task switching during flexible behavior, and specifically the suppression of erroneous movements triggered by habitual or automatic processes[7, 8, 15, 30]. However, absolute saccade initiation signals may be attributable to structures outside the BG that also converge at the SC (e.g., frontal eye field, supplementary eye field)[33, 35], because saccade latencies

after STN or SNr stimulation were uniformly distributed, and not fixed with stimulation onset (Supplementary Figs. 3 and 4).

It is well established that BG modulate a variety of behaviors, via multiple pathways in recurrent loops[1, 2, 11, 36, 37]. Indirect and direct pathways have been suggested to mediate selective suppression and facilitation effects, respectively, while hyperdirect and subthalamo-pallidal pathways may mediate global effects across the retinotopic map[11, 38–40]. Here, we have proposed that these pathways may coherently change between tasks. We can speculate a potential mechanism by which this may occur through multiple lines of evidence.

STN and caudate nucleus single-unit tonic activity is high during voluntary goal-driven saccades, but low during spontaneously generated saccades in free viewing[10, 15, 41], suggesting differences in the excitatory cortical input to the BG. Electrical stimulation of the STN can increase both glutamate and GABA release to the SNr[42, 43]. However, low-current stimulation of the STN decreases SNr activity, while high-current stimulation of the STN increases SNr activity[6], consistent with the juxtaposition of subthalamo-pallidal-nigral and 'hyperdirect' subthalamo-nigral pathways proposed here (Fig. 6). In the GPe, most increase-type neurons receive glutamatergic input and facilitate saccades (e.g., suthalamo-pallidal pathway; Fig. 6c), while decrease-type neurons receive GABAergic input and mediate reflexive saccade suppression via different pathways (e.g., striato-pallidal-subthalamo pathway[40, 44]; Fig. 6f). Excitation of GPe neurons following

cortical stimulation is mediated by cortico-STN-GPe pathways[2, 5]. Based on these findings and the data reported here, the tonic weighting of facilitatory or inhibitory subthalamo-nigral and striato-nigral pathways may vary according to the level of tonic excitatory input to the STN and CD. Mechanistically one possibility is that when STN activity is high, the subthalamo-nigral pathway may be preferentially activated due to recurrent inhibitory projections within the GPe that summate only at higher frequencies, acting as a low-pass filter[45]. On the other hand, the subthalamo-pallidal-nigral pathway may be demonstrably dominant to the subthalamo-nigral pathway during a resting state, because low-intensity STN stimulation inhibits the SNr, even though the same stimulation after SNr bicuculline injection (GABA antagonist) activates the SNr[5, 6]. Altered tonic activity across BG nuclei (e.g., altering STN tonic activity changes spontaneous firing rate in BG output, and can influence behavior[2, 46]) may broadly modify the probability of a motor or decision command reaching a neural threshold in saccade initiation structures such as the SC and frontal eye fields (e.g., refs. [47–49]).

BG have been suggested to modify decision response thresholds during a speed-accuracy tradeoff, according to greater cortical activation to the striatum (i.e., GABAergic inhibition of BG output; accuracy), or to the STN (glutamatergic excitation of BG output; speed) (Parkinson's disease[50]; functional imaging[51, 52]; modeling[53]). However, this theory relies on a predominant influence of monosynaptic efferent projections from the striatum ('striatum hypothesis'[50–52]) or the STN ('STN hypothesis'[50, 52, 53]) to BG output, and not strong opposing multisynaptic pathways such as the indirect and subthalamo-pallidal-nigral BG pathways (Fig. 6)[1–3, 11, 31]. Here, we have provided evidence for concerted task-specific changes that accounts for monosynaptic as well as multisynaptic BG pathways, that may broaden this framework of BG pathways in speed-accuracy tradeoff conditions. Further work should investigate what specific differences exist between simultaneous cortico-STN and cortico-CD signaling in oculomotor control, particularly during speed-accuracy tradeoff tasks, and determine how these STN-CD differences might influence BG pathway weightings.

To test the plausibility that differences in BG firing rate can alter downstream BG pathway weightings (as hypothesized above), we created a simple spiking neural networks model (Fig. 7a; Methods section)[54, 55] according to canonical BG projection pathways (direct, indirect, hyperdirect, and subthalamo-pallidal-nigral)[1–3, 11, 31]. In this model, we calculated the change in phase angle between STN and SNr artificial signals while varying input activation to the STN and CD. This was designed to allow a direct comparison to our observed LFP results. Using estimates of STN[7, 56] and CD[8, 57, 58] activities based on previous literature in free viewing and goal-directed saccade conditions, we observed a similar phase angle difference in artificial STN-SNr signals as in LFP recordings (Fig. 7b, and see Fig. 2b). To avoid experimenter bias in parameter selection, we also tested STN-SNr phase angle across all combinations of STN and CD firing rates from 0 to 100 spikes per second, and observed that the switch in STN-SNr is robust across a broad range of STN and CD activation levels that are in the physiologic range (Fig. 7c).

Understanding how BG signals flexibly change to promote healthy behavior may help elucidate BG disorders in which BG signals are pathologically altered, such as Parkinson's disease[59] and Huntington's disease[60]. Our findings imply that some pathological deficits may broadly involve an inability to switch flexibly between inhibitory and facilitatory BG pathways. For instance, in Huntington's disease, patients exhibit dramatically increased erroneous saccades during an anti-saccade task[61, 62], suggesting a difficulty to inhibit spurious or erroneous competing

saccade motor plans, and thus a bias toward a BG unrestrained state. In Parkinson's disease, while patients may initiate goal-driven saccades on command, their reaction times are increased[14, 63, 64], and spontaneously generated (free viewing) saccades are rare (contributing to the "Parkinson mask" diagnostic criterion[65, 66]), suggesting a pathological bias toward a BG goal-driven state, as described here. Furthermore, we found that healthy STN and SNr LFP signals were coherent in the beta range, but the phase angle between structures varied according to task, leading to the testable hypothesis that phase angle is unchanging between STN and SNr LFP signals in behaving Parkinson's disease patients when off medication. Interestingly, pathological beta frequency oscillations are well implicated in Parkinsonian motor deficits (e.g., refs. [24, 26]), and a phase shift can occur between STN and GPi LFP signals in Parkinson's disease patients by artificially altering BG signaling using on- and off-medication states[28]. Altogether, a hypothetical switching deficit may be partially rebalanced by BG treatments, particularly because clinically effective STN deep brain electrical stimulation can reduce the efficacy of pathological STN afferent and efferent projections in Parkinsonian patients in favor of cortico-striatal, thalamo-cortical, and disinhibitory direct pathway signals[66].

## Methods

**Surgery and electrophysiological recordings.** All experimental procedures were conducted in accordance with the Canadian Council on Animal Care policy on the use and care of laboratory animals, and approved by the Queen's University Animal Care Committee. Briefly, two male monkeys (Macaca mulatta) weighing 14 and 10 kg, were implanted with scleral search coils, a head restraining device, and a recording chamber under gaseous isofluorane (2–2.5%) anesthesia with the analgesic buprenorphine (0.01–0.02 mg/kg i.m.)[67]. A large recording chamber (19 mm medial-lateral × 32 mm anterior-posterior, inside diameter) was placed over the left hemisphere in both monkeys to access the head and body of the caudate nucleus, the STN, and the SNr with microelectrodes[15]. To localize the STN and SNr, we mapped these and surrounding structures extensively, within the area allowed by each chamber using a grid system. The caudate nucleus and putamen[8, 64], lateral geniculate nucleus, thalamic reticular nucleus, cranial nerve III, and internal capsule were identified electrophysiologically based on stereotyped neuronal discharge characteristics, relative anatomical locations, and visual (lateral geniculate nucleus) or eye position-related responses (cranial nerve III) where appropriate, and were used as landmarks for localizing the STN and SNr. The locations of the STN and SNr were confirmed by MRI (3 T, Siemens) in monkey O, whose implant was compatible with MRI, using the caudate nucleus, putamen, thalamus, internal capsule, and red nucleus as additional landmarks (Supplementary Fig. 3a), as previously described[68]. The locations of the STN and SNr were confirmed histologically in monkey E (Supplementary Fig. 1b). Finally, the STN[5, 7, 41] and SNr[3, 17, 20] were identified electrophysiologically (Supplementary Fig. 1c, d) by their previously described neuronal discharge characteristics. In particular these were distinguishable based on baseline firing rates, as the SNr is associated with comparatively high firing rates (>50 spikes/s) compared to the STN (25–30 spikes/s)[29, 69]. In Supplementary Fig. 1c, d, different orientations are plotted for monkey E and monkey O to maximize the number of visible stimulation sites, because of differences in the recording chamber angle between monkeys with respect to the STN and SNr. These were the same animals used previously for single neuron recordings and electrical microstimulation in the caudate nucleus, with the same paradigm[8, 15, 30, 70], which allows for a within-animal comparison of caudate nucleus, STN, and SNr stimulation results. Horizontal and vertical eye positions were sampled at 1 kHz using the search coil technique.

**Behavioral paradigms.** We trained the monkeys to perform two different saccade tasks: a free viewing task (unconstrained, free reward) and a goal-driven task (interleaved reward-driven anti- and pro-saccades; Fig. 1c). The sequence of the paradigms was counterbalanced across stimulation sites. Monkeys were centered in front of a large visual screen (50° horizontal, 30° vertical) illuminated with a diffuse gray light, and black draping occluded the monkeys' view of anything in the room save the screen. The onset and end of saccades were identified by radial eye velocity criteria (threshold: 30°/s). In reporting results, 'ipsiversive' and 'contraversive' refers to behavior (i.e., saccades elicited toward either the same or opposite visual hemifield relative to the recording or stimulation site), whereas 'ipsilateral' and 'contralateral' refers to anatomy (i.e., brain regions located in either the same or opposite brain hemisphere relative to recording or stimulation site). We examined the 200 ms pre-saccadic period in both tasks, during which no visual stimuli were present at the fixation point or saccade target in either task, but cognitive demands were very different (unconstrained and free reward during free viewing; goal-directed and reward driven during pro- and anti-saccades).

The free viewing saccade task was absent of any local visual stimuli or behavioral cues, and consisted of free viewing on a blank gray screen to engage saccadic eye movements that had no explicit goal (Fig. 1c, upper panel), and were driven only by an internal command (not a visual cue). Both free viewing saccade and goal-directed anti-saccade motor commands were internally driven (see refs. [15, 33]), but free viewing saccades were not associated with explicit task instructions or goals. We focused on the comparison between free viewing and anti-saccade conditions to remove the confound of visually driven versus internally driven movements. Juice was given randomly to the monkeys during free viewing to maintain alertness, at a rate comparable with pro- and anti-saccade task conditions, but was not time-locked to any single aspect of the free viewing task. The absence of behavioral cues was designed to replicate a simple environment in which saccadic eye movements had no explicit goal, and were driven only by an internal command generated spontaneously. Through this period the monkeys made saccadic eye movements across the gray screen whenever they pleased, with "trials" defined online but oblivious to the monkey, as follows. On each "trial", we set an invisible computer-controlled window ($\pm 10°$) on the center of the screen and waited for up to 30 s for the eyes to enter the window. We analyzed the first saccades initiated at least 300 ms after eyes entered the window. We excluded saccades initiated between 0 ms to 300 ms after the eyes entered the window arbitrarily to remove step saccades that may have passed only briefly through the window. Trials were ended after a saccade was initiated or the eyes left the window (saccade trials), or 800 ms after the eyes entered the window (no saccade trials). Each trial was followed by an intertrial interval (600 ms minimum), during which the screen remained blank. On half of the trials, electrical stimulation (see below) was initiated 300 ms after eyes entered the window and lasted until the end of the trial (stimulation duration: $M = 253/238$ ms, SD $= 33/27$ ms for monkey E STN/SNr respectively; $M = 249/239$ ms, SD $= 21/24$ for monkey O STN/SNr respectively). Stimulation was not initiated (i.e., the trial was canceled) if a saccade was initiated or the eyes left the window <300 ms after the eyes entered the window. Control and stimulation trials were randomly interleaved in each block of trials. During analysis, each saccade start point was normalized to 0. Contraversive and ipsiversive free viewing saccades were defined as those initiated spontaneously with a direction of $\pm 22.5$ degrees around the horizontal meridian. To remove the potential contamination of adjacent saccade preparatory signals, we analyzed only those free viewing saccade trials in which a second saccade was not initiated within 300 ms of ending the first saccade. We display stimulation results for all saccades from all stimulation sites, to maximize transparency. However, we have also analyzed free-viewing STN and SNr stimulation effects while stringently removing all saccades with endpoints directed outside a $\pm 3°$ square bounding window centered on 12° eccentricity on the horizontal meridian. This matched anti-saccade stimulus locations and tolerance window size. We found no significant difference in stimulation latency effects for these filtered free-viewing saccades compared to the full population (two-sample Kolmogorov–Smirnov test, $p > 0.05$; STN, $n = 76$; SNr, $n = 68$).

The anti-saccade task presented a peripheral visual cue, but required the monkey to inhibit a visually guided saccade toward the cue in favor of an internally driven saccade directed to blank space 180° away[8, 18] (Fig. 1c, lower panel). Unlike the free viewing task, the anti-saccade task required the monkey to follow explicit task instructions, and to initiate the internally-driven saccade toward a rewarded goal. Anti-saccade trials were randomly interleaved with visually guided pro-saccade trials (i.e., look toward the visual cue), in order to maintain active engagement in the task (Fig. 1c, Supplementary Figs. 2, 4 and 5). Each trial was preceded by a 600 ms intertrial interval during which the screen was illuminated with a diffuse light. After the removal of the background light, a fixation point appeared in the center of the screen, and the monkeys were required to direct eyes toward the fixation point within 30 s. After they maintained fixation for 900–1200 ms, a red stimulus was presented either 12° left or right from the fixation point and the monkeys generated a saccade either toward the stimulus (pro-saccade trial) or to the opposite direction of the stimulus (anti-saccade trial) within 600 ms based upon fixation point color (red: pro; green: anti). Trials with saccade latencies (defined as the delay between eccentric stimulus appearance and saccade onset) below 70 ms were excluded online because they are associated with a 50% probability of being correct, reflecting anticipatory responses[18]. We examined horizontal saccades, because unilateral SNr stimulation skews vertical saccades horizontally[16], and visual receptive fields for STN neurons are very large and concentrated in the contralateral hemifield[7, 41]. The trial instruction (pro/anti) was indicated by the color of the fixation point when it appeared. After making a saccade to the appropriate location, the monkey was required to maintain fixation for 150–350 ms (on the peripheral red stimulus on pro-saccade trials, or on a peripheral invisible window of blank space in the mirror position of the peripheral red stimulus after saccade onset on anti-saccade trials). The monkeys received a liquid reward after each correct performance. A 200 ms gap was introduced before stimulus appearance during which the fixation point disappeared and the monkeys maintained fixation on the blank screen. We adopted this temporal gap to reduce active fixation signals during saccade preparation[71], increasing the probability of eliciting behavioral effects by low-current electrical stimulation. During the electrical stimulation experiment (see below), stimulation was delivered from stimulus appearance until saccade initiation on half of the trials (stimulation duration: $M = 234/185$ ms, SD $= 88/75$ ms for monkey E STN/SNr respectively; $M = 265/264$ ms, SD $= 65/75$ for monkey O STN/SNr respectively). The pro/anti instructions, left/right stimulus locations and stimulation/control trials were randomly interleaved in each task block. We excluded any stimulation site with <240 correct trials (i.e., 30 correct trials per task condition).

**Local field potential recording**. LFP was recorded in the STN and the SNr simultaneously, via pairs of monopolar tungsten microelectrodes (impedance: 0.1–1 MΩ, Frederick Haer). For each recording session, we identified both the STN and SNr electrophysiologically, and then selected pairs of STN-SNr LFP recording locations based on combinations of those sites where we encountered task-related activity, broadly defined as any neuron (multi- or single-unit) that increased or decreased firing rate depending on task conditions after stimulus onset, as described previously in the caudate nucleus in the same monkeys[8, 30]. The minimum distance between recording sites within each structure was 300 μm. At each STN-SNr recording site pair, the monkey performed from 500 to 2000 interleaved pro- and anti-saccade trials and 500–2000 free viewing saccade trials. Raw LFP data were band pass filtered between 0.5 and 60 Hz (Butterworth filter, 2nd order), and LFP data were analyzed within sites. We examined STN and SNr signals using coherence and physiologically generated phase angle differences (Figs. 1a, 2, and 7b, c). Coherence is a measure of the variability of time differences between signals (i.e., phase locking)[23], and the time difference between signals is represented by phase angle. Coherence approaches 1 if the phase angle is stable and constant over time between signals, while coherence approaches zero if the phase angle between two signals varies frequently.

**Electrical stimulation parameters**. Constant-current charge-balanced biphasic pulses (anode first, 500 μs pulse width, 10–40 μA, 100 Hz, behaviorally contingent durations as defined above) were delivered to the STN or to the SNr via a monopolar tungsten microelectrode (impedance: 0.1–1 MΩ, Frederick Haer) using a stimulator (Grass S88, Grass Tech) attached to a pair of constant current stimulus isolation units (Grass PSIU6). We chose the stimulation parameters based on previous reports[15, 72] to preferentially activate gray matter structures, and reduce current spread to the internal capsule (i.e., low current, and long pulse width)[73]. Electrical current was measured by the voltage drop across a 1 kΩ resistor in series with the return lead of the stimulator. For each penetration, we first identified the STN or SNr electrophysiologically and then stimulated at sites evenly along the penetration at 500 μm intervals. We confirmed similar results in a subset of stimulation sites where we encountered task-related neurons, broadly defined as any neuron that increased or decreased firing rate depending on task conditions after stimulus onset, as described previously in the caudate nucleus in the same monkeys[8, 30]. Eleven stimulation sites were removed from analysis, because they were located at the border of the STN and SNr, and may have exhibited characteristics of both STN and SNr stimulation. At higher currents (30–40 μA), some stimulation sites in the STN or the SNr induced saccades such that saccade vector endpoints were very tightly clustered in the free viewing task, and the monkey was only able to produce saccades toward one hemified in the goal-directed saccade task regardless of task instruction. At these sites, induced saccadic reaction times had a mean of 177 ms ± 93 ms in the STN, and 173 ± 24 ms in the SNr, which is substantially longer than the evoked saccadic reaction times associated with stimulation of the frontal eye field or SC (~50 and 30 ms respectively[74, 75]), suggesting that antidromic activation of either structure was unlikely. In addition, stimulation in SNr penetrations (approximately the same distance from the internal capsule as the STN) did not cause a contraversive movement bias (see Results), as predicted by current spread to the internal capsule[76], and stimulation within the internal capsule superior to the STN did not reveal consistent behavioral effects as stimulation within the STN (not shown). Based on this, and because STN stimulation can both activate and suppress SNr neuronal activity[6, 42, 43], our stimulation effects in the STN and SNr are unlikely the result of current spread to other brain areas, particularly the frontal eye field or SC. Nevertheless, the possibility cannot be completely discounted, and at these sites the current amplitude was lowered below 40 μA in order to record enough of both ipsiversive and contraversive correct goal-directed trials for the analyses of reaction times.

**Analysis of saccade direction bias**. To quantify the effect of electrical stimulation on saccade direction during free viewing, we calculated the following index for each stimulation site:

$$Saccade\ direction\ bias = S_{bias} - C_{bias} \quad (1)$$

$$S_{bias} = \frac{\#\ contraversive\ stimulation\ trial\ saccades\ -\ \#\ ipsiversive\ stimulation\ trial\ saccades}{total\ \#\ stimulation\ trial\ saccades} \quad (2)$$

$$C_{bias} = \frac{\#\ contraversive\ control\ trial\ saccades\ -\ \#\ ipsiversive\ control\ trial\ saccades}{total\ \#\ control\ trial\ saccades} \quad (3)$$

where $S_{bias}$ and $C_{bias}$ denote stimulation and control trials, respectively. The value of this index is close to $\pm 2$ if there is a large difference between the saccade direction ratio on stimulation and control trials, while it is close to zero when the

difference between stimulation and control trials is negligible. Horizontal and vertical components of saccade direction were isolated and analyzed individually using Eqs. (1–3). Positive indices indicate that stimulation caused a rightward horizontal saccade direction bias, or an upward vertical saccade direction bias. Negative indices indicate that stimulation caused a leftward horizontal saccade direction bias, or a downward vertical saccade direction bias.

**Saccade latency and frequency**. Saccade latency and frequency were examined to determine the mechanism of saccade bias (e.g., contraversive facilitation, or ipsiversive inhibition, etc). In the free viewing task, saccade latency was defined as the fixation duration from electrical stimulation onset time to the initiation of the first saccade in each trial. In goal-driven trials, saccade latency was defined as the fixation duration from peripheral visual stimulus onset (i.e., also from electrical stimulation onset time) to saccade onset. To quantify the effect of electrical stimulation on saccade latency, we calculated the following index for each stimulation site (derived from[8]).

$$Saccade\ latency\ index\ = \frac{C_{latency} - S_{latency}}{|C_{latency} - S_{latency}| + 2RMSerror} \quad (4)$$

$$RMSerror = \sqrt{SSE/(N-2)} \quad (5)$$

where $S_{latency}$ and $C_{latency}$ indicate average saccade latency in stimulation and control trials, respectively. RMSerror was calculated using Eq. (5). SSE is the squared sum error around the averages on stimulation and control trials. $N$ indicates the total number of trials. This index is close to $\pm 1$ if the difference between average saccade latencies during control and stimulation trials is much larger than the variance in saccade latencies, and is close to zero when the difference between average saccade latencies is negligible compared to the variance. Positive and negative indices indicate that stimulation shortened and prolonged saccade latencies, respectively. This is a more conservative index of change in saccade latency, which negates possible trends due to non-meaningful variance (noise) in the data. The investigator could not be blinded to the task condition in this study during data collection or analysis.

**BG spiking neural networks model**. A simple spiking neural networks BG model was created to estimate whether altering the input activation of the STN and CD could result in an STN-SNr phase angle shift. Each nucleus ("node") contained 1000 artificial neurons (McCulloch-Pitts neuron model[54, 55]), which projected to other nuclei according to major BG projection pathways[1–3, 11, 31]. BG activity in all relay (GPe, relay stage of STN) and output structures (SNr) were treated as a closed system, such that they did not receive any manual or pre-determined tonic activation input. We applied external excitatory inputs to the STN and CD only (BG input nuclei), conceptually representing cortical input activity. STN and CD cortical input was specified as a firing rate (spikes/s), and cortico-STN and cortico-CD input firing rates were adjusted according to previously reported task-dependent STN and CD activities, while examining differences in STN-SNr phase angle. Independent of firing rate, different oscillatory frequencies were introduced into the STN and CD (15 Hz, CD; 45 Hz, STN, for the results presented here), in order to differentiate STN and CD signals in coherence and phase analyses. However, the specific oscillatory frequency employed in the STN and CD was used for measuring purposes only, and selecting different values did not alter BG model behavior (because spike counts were adjusted to keep firing rates constant across frequencies). Neuronal activities of other BG nodes were determined exclusively by signal transmission from the STN and CD through the canonical direct/indirect/hyperdirect/subthalamo-pallidal-nigral pathways. Anatomical localizations of projection neurons within each nucleus were assigned randomly. The architecture of efferent signals between BG nuclei is described in Fig. 7a. Neurons targeted by CD, STN, and GPe signals each received projections from five, ten, and five random projection neurons, respectively, based on widespread anatomical distributions of STN efferents on downstream nuclei[38, 77]. Neuronal projection targets were randomly assigned during model initialization, but remained constant when testing between task conditions as well as across artificial recording sessions. Each artificial neuron signaled a binary output state (0, no activation; or 1, 'action potential'), and this output was excitatory (+1) or inhibitory (−1) to downstream neurons according to the source BG nucleus identity (CD, inhibitory; STN, excitatory; GPe output node, inhibitory; GPe recurrent node, inhibitory). A neuronal output state of 1 (action potential) or 0 (no activation) was determined at each time point for each neuron, according to a threshold level of average input activation of greater than or less than 0.5 to that neuron, respectively. This model was intended only to test the plausibility of a subthalamo-nigral phase angle change in a simple BG framework; the interested reader may refer to more complex models of BG in functional behavior or decision making (for review, refs. [9, 57, 78, 79]).

**Code availability**. Code for the BG neural networks model (MATLAB, MathWorks, Natick MA) is available from the authors upon reasonable request.

**Data availability**. All relevant data are available from the authors upon reasonable request.

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

## Acknowledgements

We thank Ann Lablans, Mike Lewis, and Sean Hickman for outstanding technical assistance, and members of the Munoz lab for feedback on an earlier version of the manuscript. This work was supported by the Canadian Institutes of Health Research Foundation Grant #148418 to D.P.M.; J.J.J. was supported by a Canadian Institutes of Health Research Vanier Canada Graduate Scholarship. D.P.M. was supported by the Canada Research Chair Program.

## Author contributions

J.J.J., M.W. and D.P.M. conceived the study and analyses. J.J.J. collected the initial data set, wrote the analysis software, performed all analyses, wrote the first draft, designed and programmed the neural networks model, and assembled the paper. R.L. and D.P.M. provided training, advice, and critique of paper.

## Additional information

**Competing interests:** The authors declare no competing financial interests.

