## [Peer Review File · Nature Communications]

Reviewers' comments:

Reviewer #1 (Remarks to the Author):

The basal ganglia (BG) are consisted of multiple parallel pathways from the input to output channels. However, how these circuits function in coherence to achieve a certain goal has been unclear. By the measurement of neuronal activity and electrical stimulation in the STN and SNR, this study presents evidence suggesting that the tonic weighting between inhibitory and excitatory pathways from the STN to the SNR may alter according to behavioral context.

First, to distinguish which pathways (STN-GPE-SNR vs. STN-SNR) are detected, they analyzed phase difference and the strength of coherence between LFP signals from the STN and SNR. The results showed that STN-GPE-SNR pathway was dominant for the free-viewing task and STN-SNR pathway was dominant for the anti-saccade task.

They next performed unilateral electrical micro-stimulation in the STN and SNR. SNR stimulation produced largely suppressive effects in both tasks. However, STN stimulation produced clear task specific effects indicating different source of information depending on the task. They suggest that, during free viewing, the facilitatory control such as the STN-GPE-SNR and the STR-SNR pathways are dominant, consistent with fast automatic or sensory-driven movements toward unexpected stimuli. In contrast, during a goal-driven condition, inhibitory control such as caudate-GPE-STN-SNR pathway is dominant, consistent with a reduction of erroneous movements in favor of voluntary motor commands with strong preparatory activity to elicit a saccade accurately.

The idea of switching between relative strength of STN-SNR, STN-GPE-SNR, and CD-GPE-SNR is interesting and novel. Although what electrical stimulation exactly does is not clear, it still gives us strong evidence that there are shift between pathways. I suggest major revisions mainly in terms of presentation to make the report clear to the readers.

Major

1) The terminology used in the present paper is very confusing. In particular, because we are so used to use 'direct' pathway to describe the striatum-GPI/SNR pathway, it is better not to use 'direct pathway' for STN-SNR. The schemes (Fig1, 2, 7) are also confusing partly because of the terminology above. I suggest to use the figure like Jahanshahi (Nature review neuroscience 2015) Figure 2 at the beginning of the paper.

2) It is not clear how long/often/when the stimulation was applied in the free viewing task. It said (L129) 'it began 300ms after the monkey fixated near the center of the screen, and ended when a saccade was detected or after 800ms'. Does this mean stimulation may happen 'after saccade'? If so, were there any plastic changes – gradual changes in saccadic parameters after several stimulations or even after stimulation stopped? (see Williams and Eskandar, Natureneurosci (2006), Nakamura and Hikosaka J.Neurosci (2006)).

3) In Fig7, the effect of cortical inputs to STN is ignored. As the caudate and STN also receive cortical inputs, the discussion on the differential (or common) inputs from the cortex should be discussed.

Minor

1) The time sequence of tasks events, eye movements, LFP recording period, and the stimulation period should be explicitly shown in figures. For example, it went 'no visual stimuli were present during 200ms pre-saccade period (L417)'. Does it mean, in the anti-saccade task, there was a gap between fixation offset and target onset? Also, was the stimulation applied before and also, after

saccades?

2) L138 It said $n=47/68$ which is inconsistent with Table 1(Supp).

3) L419 The parameters of electrical stimulation were determined based on the previous reports. Nevertheless, if the authors tried different parameters, it would be beneficial show how parameters affect the results. Especially, DBS therapy in the STN is conducted, usually by higher frequencies.

Reviewer #2 (Remarks to the Author):

In daily life, we sometimes need to facilitate motor action but sometimes need to suppress the same action dependent on context. The basal ganglia have multiple neural pathways that either facilitate or suppress body movements. How the basal ganglia pathways control our behavior in adaptive manners is a big issue.

In this study, the authors found that the basal ganglia adaptively change the weightings of the subthalamo-nigral pathway (STN-SNr) and the subthalamo-pallidal-nigral pathway (STN-GPe-SNr) across two tasks (i.e., free viewing task in which eye movements were unconstrained and monkeys obtained a free reward, and anti-saccade task in which the monkeys were required to make a saccade to the opposite direction of a peripheral visual cue). They first recorded LFP simultaneously from the subthalamic nucleus (STN) and the substantia nigra pars reticulata (SNr). They found that the phase angle of their LFPs changes between the tasks, suggesting that the weightings of the STN-SNr and STN-GPe-SNr pathways are changed dependent on the tasks. Second, they electrically stimulated the STN and examined the effect of the stimulation on eye movements during the tasks. They found that the effect of the STN stimulation is opposite between the tasks. This result suggests that the weighting of the STN-GPe-SNr pathway is dominant during the free viewing task and the weighting of the STN-SNr pathway is dominant during the anti-saccade task. These two experiments seem to suggest that the basal ganglia adaptively change the weighting of each pathway dependent on task demands. Their idea and findings are new and important to understand how the basal ganglia adaptively control motor action. But, I have some concerns about how to interpret their data.

Major concerns:

(1) They recorded LFP simultaneously from the STN and SNr. The phase angle of their LFPs was about 0 degree or slightly negative during the free viewing task. On the other hand, the phase angle was about 10 degree during the anti-saccade task. The authors explained that the increase in the phase angle occurs if the STN-SNr pathway becomes more dominant than the STN-GPe-SNr pathway. However, the increase was only 10 degree. As the authors predicted in the earlier part of the manuscript, the phase angle increase should be close to 180 degree if the STN-SNr pathway becomes more dominant. The authors discussed that the phase angle only slightly increased because of the effects of other pathways. But, I don't understand why other pathway can interrupt the full increase in the phase angle. A mathematical simulation using a simple neural network model of basal ganglia pathways could be useful for readers to understand why the increase in the phase angle was so small.

(2) The authors showed the effect of electrical stimulation on saccade latency in "pro-saccade task" in Supplementary Materials (Figure S6). This is very important information. But the authors did not explain this data in the main manuscript. Furthermore, curiously, the effect of STN stimulation was the same (i.e., inhibitory) between the anti- and pro-saccade tasks. Although the pro-saccade task just required to execute a saccade to a visual target, anti-saccade task needs to a process that inhibits a

saccade to a visual target. Therefore, it is reasonable that the STN-SNr pathway, which inhibits motor action, becomes more dominant during the anti-saccade task. Why did this pathway become more dominant even during the pro-saccade task? Author should show the results of the pro-saccade task in the main manuscript, and should explain why.

Other concerns:

(3) The authors showed the STN-SNr LFP coherence in Figure 3a. I want to see the “time course” of the coherence that is aligned by saccade onset. Heat map that includes all frequency bands would be nice.

(4) The authors showed the phase angle before saccade onset. I want to see the phase angle around and after saccade onset, as well. The authors argued that the effects are tonic. If so, the same phenomenon should occur during other timings.

Reviewer #3 (Remarks to the Author):

This manuscript describes a study of signalling pathways in the basal ganglia (STN and SNr) of monkeys, comparing between a free viewing saccade task and an instructed anti-saccade task. The hypothesis is that the relative strength of functionally opposing BG pathways – the inhibitory STN-SNr pathway and the facilitatory STN-GPe-SNr pathway – are modulated based on task demands. Two approaches are used: First, the authors examine coherence and phase shift between pairs of simultaneous LFP recordings in STN and SNr, and find a phase shift difference of beta oscillations between free viewing and anti-saccade conditions. Second, the authors examined the effects of microstimulation in STN and SNr on the latency and hemifield bias of saccades. They found that while stimulation of SNr always inhibited saccades (contra or bilaterally, depending on site), stimulation of STN was task-dependent: facilitating contraversive saccades during the free viewing condition and inhibiting them during anti-saccade trials. Both approaches yield congruent conclusions, which support the hypothesis that the STN-SNr pathway dominates during the goal-driven saccades while the STN-GPe-SNr pathway dominates during free viewing.

Overall, this is a very solid study that provides important new information for making sense of basal ganglia circuitry. The study is well done and the paper is well organized, although there are some typos and missing information here and there, as well as some awkward writing. Experimental procedures are well planned and more generally, the study is technically sound. Statistical analyses are appropriately done and reported, though the authors’ choice of different statistical tests is unclear and needs more justifications. I found the stimulation results much more compelling than the LFP analyses, but both are complementary.

A general critique is that this study does not really address “task-dependence” in the broad sense, but instead addresses the more specific issue of “inhibition” of automatic responses (clearly an important strategy to prevent premature responses in anti-saccade tasks). I would suggest the authors frame their results in terms of inhibition instead of in a way that appears to pertain to task-dependence in general, especially since the role of the STN in control of inhibition has already been well-established. The major difference between the tasks is that the anti-saccade task demands the withholding of immediate responses, engaging deliberate inhibition. So the difference is not a general “task-dependence”, but a specific difference in inhibitory control that the monkey learned through training. In fact, I’d recommend making this clear throughout by describing those trials as “anti-saccade” instead of “goal-directed”, in all figures and text.

Furthermore, for this reason it would be useful to look at pro-saccades. Since they are interleaved among anti-saccades they might also be more inhibited than pure free viewing. Indeed, STN-SNr coherence in pro-saccades (Fig S4) looks pretty similar to coherence in anti-saccades (Fig S3). But what were the effects of SNr and STN stimulation during the pro-saccade trials? How did they differ from the anti-saccade condition, and do they make sense in the context of the scheme proposed in Figure 7? I agree with the authors' motivation for focusing on anti-saccades because of the absence of a visual stimulus at the saccade target site, but looking at the pro-saccade data might nevertheless be useful. After the fixation color appears, and indicates a pro-saccade trial, the inhibitory control might be relaxed a bit (like the free-viewing trials) but nevertheless the saccade is still goal-directed (like anti-saccade trials). It might be useful to look at LFP coherence and phase during the fixation period, prior to cue appearance.

The authors predict a 180 deg STN-SNr phase difference between the free-viewing and the anti-saccade condition. If I understand it correctly, this is because of an expected change in sign when the STN-GPe-SNr pathway is stronger than the STN-SNr pathway. But that's not what is seen. The phase difference is much smaller and rather weak. For most sites (except two: Fig 2b), the phase shifts are distributed around 0. How do the authors explain this discrepancy? To me, it seems that phase shifts could be produced by changes in sign, or by changes in transmission delays, or both – making interpretation challenging. My understanding is that beta oscillations are a product of the entire BG loop through cortex and thalamus, so I would have predicted only slight phase shifts depending on whether the signal gets through the STN-SNr or the (presumably slightly slower) STN-GPe-SNr pathway. A 180 deg shift would be difficult to imagine (it would correspond to a delay of ~25ms). Perhaps I don't understand the logic behind the authors' interpretation of the phase shifts, in which case I'd appreciate more clarification.

The authors discuss their results in terms of the "comparative weighting of BG pathways" (e.g. line 5), but that phrase makes it sound like a hypothesis about changes in synaptic connectivity. As far as I can tell, the authors are really suggesting that during anti-saccade trials, there is more inhibition of GPe by the caudate, reducing the possibility for STN to inhibit the SNr and instead allowing its facilitatory pathway to dominate. Is that right or are the authors trying to argue for something more?

SPECIFIC COMMENTS

Isoda & Hikosaka (2008) already made a good case that the STN is involved in task switching. That paper is cited in the methods, but it seems relevant for the introduction and discussion as well. The numbers of sites where a significant effect of SNr stimulation has been found are confusing. For instance, in line 138 it is stated that out of 68 sites, 47 produced only contraversive inhibitions. But in line 162, it is stated that stimulation significantly affected contraversive saccade initiation in 18 sites. This should be clarified.

It's not clear how the authors determine online, for the microstimulation, the 200ms pre-saccadic period in the free-viewing task.

How did the authors choose between stimulation intensities (between the 10-40 μ A ranges)?

It is not clear if the authors excluded free saccades that did not match the spatial characteristics of the ones studied in the anti-saccade task. How large was the tolerance window in the anti-saccade task, and was it comparable to the saccades examined in free-viewing?

It is not clear how the authors choose their statistical tests to evaluate the effect of stimulation on saccade parameters. A t-test is used to assess the effect of microstimulation on saccade endpoint distributions, but a KS-test for the effect of stimulation on saccade frequency. More problematically, the KS-test is sometimes used to test the effect of stimulation on saccade latencies (line 173-174),

but most of the time the authors use t-tests (for example in line 158). Why??

Talking about “saccade latency” in the free-viewing condition is rather strange, since there is no known trigger with which to compare saccade onset. The authors instead use the start of the last central fixation – but then the quantity being measure is actually fixation duration. I understand the desire to compare these to the latencies in the anti-saccade task, but it would be important to acknowledge the differences between measures and potential confounds those differences could introduce.

Lines 22-23: “direct” and “indirect” should be reversed.

Line 95: typo: glutamatergic instead of glutamatertic.

Line 105: Why is it $p < 0.05$ for a 99% confidence interval?

The scale of the phase variable in Figure 3a could be more informative. In particular, it's important to know where the value of 0 lies (it doesn't even appear to correspond to a tick-mark).

Line 180: I would replace the title of the section with: Effect of STN stimulation was task dependent

Line 256-257: Should be “subthalamo-pallido-nigral” pathway

Line 312: Don't you mean “STN stimulation”?

Lines 331-332: “while hyperdirect and STN-GPe-SNr pathways may mediate global effects.” This part of the sentence is not explicit enough. The authors should at least briefly describe some of these “global” effects.

When comparing the results of the present experiment with those on the caudate published previously (lines 311-313 of the discussion), did the authors notice a difference of effect strength (quantitatively) between the stimulation of the caudate and the STN in the free-viewing task? Both induce a contraversive bias but I was wondering if the weighting between the classic direct pathway (CD-SNr) and the STN-GPe-SNr pathway is different.

Are the pairs shown in Figures S2-4 the same in each corresponding panel? If so, then it could be useful to overlay the three conditions (each with different colors) in the same figure, and discuss the differences.

We thank the reviewers and editors for these valuable comments on our original manuscript entitled “Evidence for a task-dependent switch in subthalamo-nigral basal ganglia signaling” submitted to *Nature Communications*. The manuscript has been improved significantly by taking into account all of the suggestions. In particular, we have created and included results from a new spiking neural networks model of the basal ganglia (including hyperdirect, direct, indirect, and subthalamo-pallido-nigral pathways). This replicated the observed task-dependent phase shift in STN-SNr LFP, across a broad range of model input parameters, and has added strong support to our hypothesis of the underlying mechanism of task-dependent pathway weightings in the BG. We have also described in more detail STN, SNr, and CD results collected during pro-saccades. This revised manuscript therefore includes converging evidence from neuronal recording; electrical stimulation; and spiking neural networks modeling; collected in three basal ganglia nuclei, and within three different behavioral conditions. We believe the manuscript has been substantially improved and we hope that it is now acceptable for publication in *Nature Communications*.

In the following sections, red sentences indicate the comments of the reviewers and black sentences indicate our point-by-point replies to these comments. Revisions made in the main text are recorded by tracked changes, and marked in red.

Reviewer #1 (Remarks to the Author):

The basal ganglia (BG) are consisted of multiple parallel pathways from the input to output channels. However, how these circuits function in coherence to achieve a certain goal has been unclear. By the measurement of neuronal activity and electrical stimulation in the STN and SNR, this study presents evidence suggesting that the tonic weighting between inhibitory and excitatory pathways from the STN to the SNR may alter according to behavioral context.

First, to distinguish which pathways (STN-GPE-SNR vs. STN-SNR) are detected, they analyzed phase difference and the strength of coherence between LFP signals from the STN and SNR. The results showed that STN-GPE-SNR pathway was dominant for the free-viewing task and STN-SNR pathway was dominant for the anti-saccade task.

They next performed unilateral electrical micro-stimulation in the STN and SNR. SNR stimulation produced largely suppressive effects in both tasks. However, STN stimulation produced clear task specific effects indicating different source of information depending on the task. They suggest that, during free viewing, the facilitatory control such as the STN-GPE-SNR and the STR-SNR pathways are dominant, consistent with fast automatic or sensory-driven movements toward unexpected stimuli. In contrast, during a goal-driven condition, inhibitory control such as caudate-GPE-STN-SNR pathway is dominant, consistent with a reduction of erroneous movements in favor of voluntary motor commands with strong preparatory activity to elicit a saccade accurately.

The idea of switching between relative strength of STN-SNR, STN-GPE-SNR, and CD-GPE-SNR is interesting and novel. Although what electrical stimulation exactly does is not clear, it still gives us strong evidence that there are shift between pathways. I suggest major revisions mainly in terms of presentation to make the report clear to the readers.

We thank the Reviewer for these positive comments, and for their helpful comments below.

Major

1) The terminology used in the present paper is very confusing. In particular, because we are so used to use ‘direct’ pathway to describe the striatum-GPI/SNr pathway, it is better not to use ‘direct pathway’ for STN-SNr. The schemes (Fig1, 2, 7) are also confusing partly because of the terminology above. I suggest to use the figure like Jahanshahi (Nature review neuroscience 2015) Figure 2 at the beginning of the paper.

We thank the Reviewer for these comments, and apologize for any confusion. We fully agree that ‘direct pathway’ is improper to describe the monosynaptic STN-SNr pathway. However, we carefully read through the manuscript (and searched the text for every use of the term ‘direct’), and were unable to find any instance in the manuscript when ‘direct’ was used to describe any STN-SNr pathway. As first described in the Introduction (Lines 24-25) and in Fig 1, throughout the manuscript we attempted to use the convention of ‘direct’ for CD-SNr, ‘indirect’ for CD-GPe-STN-SNr, ‘hyperdirect’ for STN-SNr, and ‘subthalamo-pallidal’ or ‘subthalamo-pallido-nigral’ for STN-GPe-SNr pathways. This terminology reflects the canonical anatomy of oculomotor BG pathways, and as noted by the Reviewer, is well used previously (e.g., Hikosaka O, Takikawa Y, Kawagoe R [2000] Role of the basal ganglia in the control of purposive saccadic eye movements. *Physiol Rev.*; Nambu A, Tokuno H, Takada M [2002] Functional significance of the cortico-subthalamo-pallidal ‘hyperdirect’ pathway. *Neurosci Res.*).

We agree with the Reviewer that the schematics in Fig 1 and Fig 2 were somewhat confusing and redundant. Therefore, we have refined Fig 1 and Fig 2 into a single descriptive figure (new Figure 1). We have also included a schematic describing the BG pathways of the new spiking neural networks model (Figure 7a). We believe that this significantly clarifies the hypothesis and methodology of the manuscript. We agree with the Reviewer that Fig 2 from Jahanshahi et al. (2015) is clear. In fact, our BG schematics were modeled in large part after Jahanshahi et al. (2015), as well as after the extremely well-known diagrams from Okihide Hikosaka and Atsushi Nambu (e.g., Hikosaka et al., 2000; Nambu et al., 2002). We created several drafts of new BG schematics to attempt to more closely approximate Jahanshahi et al., to fully explore the Reviewer’s comment. However, when we presented these options internally and in consultation with external scientists, the feedback was highly positive toward the simplified BG circuit diagram. Otherwise, we were told that the figures included an overwhelming number of nuclei and projections. Therefore, although we have extensively re-designed Fig 1, Fig 2, and Fig 7 according to the Reviewer’s suggestions, we would prefer to maintain the layout of BG nuclei within these figures.

2) It is not clear how long/often/when the stimulation was applied in the free viewing task. It said (L129) ‘it began 300ms after the monkey fixated near the center of the screen, and ended when a saccade was detected or after 800ms’. Does this mean stimulation may happen ‘after saccade’? If so, were there any plastic changes – gradual changes in saccadic parameters after several stimulations or even after stimulation stopped? (see Williams and Eskandar, *Natureneurosci* (2006), Nakamura and Hikosaka *J.Neurosci* (2006)).

We thank the Reviewer for the opportunity to clarify this question. Saccades were detected online during free viewing, using velocity and displacement threshold criteria. Stimulation was not applied

between detected saccade onset and saccade offset periods. This prevented stimulation from being initiated while a saccade was in progress, and also cancelled the stimulation if a saccade was initiated. We have clarified this point in the Methods (Line 468-470) to address the Reviewer's concern.

To further address the Reviewer's concern, we dichotomized each free viewing file by splitting them according to their temporal midpoint. We compared populations containing mean free viewing saccadic latency (as defined in the manuscript) between early and late stimulation trials, and found no significant difference (paired t -test, $p > 0.05$).

Because 1) stimulation was only applied during gaze fixation; 2) we used a short periodic stimulation with low current and low frequency parameters (e.g., Ranck 1975); and 3) we found no significant difference in saccade latencies between stimulation trials occurring early and late in a recording session, we conclude that there were no plastic changes in saccade parameters due to stimulation that would confound our results. It is also worth mentioning that because our free viewing task did not contain any visual cues (Williams and Eskandar, 2006) or reward corresponding to any behavioral event (Nakamura and Hikosaka, 2006), there was arguably no opportunity for learning or reward-based changes as described in these papers. Therefore, we do not wish to claim that plastic changes of saccadic parameters cannot occur via STN or SNr stimulation, but simply that we did not have a task design that would prompt or allow us to test whether the results that Williams and Eskandar (2006) and Nakamura and Hikosaka (2006) described in the caudate nucleus can occur in the STN and SNr.

3) In Fig7, the effect of cortical inputs to STN is ignored. As the caudate and STN also receive cortical inputs, the discussion on the differential (or common) inputs from the cortex should be discussed.

We thank the Reviewer for this comment. It was not our intent to ignore the effect of cortical inputs to the STN or the CD. In fact, differences in cortico-basal ganglia input activity between free viewing and goal-directed behavioral conditions is a keystone of our hypothesized mechanism of action (Discussion, Lines 352-369). The schematic in old Figure 7 (now Figure 6 in the revised manuscript) was specifically intended to summarize the stimulation effects of the SNr, STN, and CD, and to illustrate the hypothesized difference in BG signalling pathways based on our results. To highlight this, at each stage of stimulation in the BG (i.e., SNr/STN/CD) we presented only the downstream BG nuclei directly comprising the pathways underlying our hypothesized results (i.e., direct/indirect/hyperdirect/subthalamo-pallidal pathways).

With respect to differences in the anatomical distribution of cortical efferents to the BG, we acknowledge there could certainly be differences in input signals to the STN and CD. However, we are concerned this broad topic cannot be properly addressed given the word limits in the manuscript as the Reviewer has asked, and furthermore is outside the direct scope of our results. We are also unaware of any studies directly comparing cortico-STN and cortico-CD signals during saccade behavior. Therefore, in this manuscript we would like to emphasize the existence of a task-dependent switch in signaling pathways *within* the BG. This conclusion does not preclude – and could conceivably be facilitated by – differences in the anatomical distribution of cortical efferents to the STN versus the CD.

To address the Reviewer's concern within the limitations and scope of this manuscript, we have included a caveat in the Discussion (Lines 372-374) stating that it is feasible that there are differences between cortico-STN and cortico-CD input signals which cannot be addressed in the current manuscript, and these should be investigated in future recording studies.

Minor

1) The time sequence of tasks events, eye movements, LFP recording period, and the stimulation period should be explicitly shown in figures. For example, it went ‘no visual stimuli were present during 200ms pre-saccade period (L417)’. Does it mean, in the anti-saccade task, there was a gap between fixation offset and target onset? Also, was the stimulation applied before and also, after saccades?

We thank the Reviewer for this comment. On Line 417 (Line 447 on the revised manuscript), we stated “We examined the 200 ms pre-saccadic period in both tasks, during which no visual stimuli were present at the fixation point or saccade target in either task, but cognitive demands were very different...”. The Reviewer is correct: as described in the Materials and Methods Line 502-504, we used an anti-saccade gap task in which there was a 200ms gap between fixation offset and target onset. Therefore at the time of stimulation onset in the anti-saccade task, the fixation point was no longer visible (i.e., gap task), and there was no visual stimulus presented at the saccade target (i.e., anti-saccade task).

As described in the Materials and Methods Line 509-513, stimulation was applied from stimulus appearance until saccade initiation (or the eyes left the fixation bounding window). Therefore stimulation was applied before, and not after, saccade initiation.

To clarify this, as the Reviewer has requested we have created a figure that illustrates the time sequence of task events, eye movements, LFP recording period, and stimulation period (Supplementary Figure 1).

2) L138 It said $n=47/68$ which is inconsistent with Table 1(Supp).

We gratefully thank the Reviewer for identifying this ambiguity. To be conservative, we show data from all sites that did not exhibit bilateral inhibition ($n=47/68$), we did not specifically pick only those exhibiting a contraversive bias ($n=35/68$). All statistical analyses reported in this section also reflected this combined population, as a more conservative and informative analysis. We have clarified which population of STN stimulation sites was analyzed in the Results (Line 145-146).

3) L419 The parameters of electrical stimulation were determined based on the previous reports. Nevertheless, if the authors tried different parameters, it would be beneficial show how parameters affect the results. Especially, DBS therapy in the STN is conducted, usually by higher frequencies.

We thank the Reviewer for this comment, and agree that this is an interesting question. Unfortunately, all stimulation sites were collected with the parameters reported, so we cannot address this question with the current data. During this experiment, our focus was on reducing the possibility of current spread, whereas DBS therapy in the STN is of course focused on clinical behavioral improvement rather than mechanism of action. This would be an excellent topic for a future study.

Reviewer #2 (Remarks to the Author):

In daily life, we sometimes need to facilitate motor action but sometimes need to suppress the same action dependent on context. The basal ganglia have multiple neural pathways that either facilitate or suppress body movements. How the basal ganglia pathways control our behavior in adaptive manners is a big issue.

In this study, the authors found that the basal ganglia adaptively change the weightings of the subthalamo-nigral pathway (STN-SNr) and the subthalamo-pallidal-nigral pathway (STN-GPe-SNr) across two tasks (i.e., free viewing task in which eye movements were unconstrained and monkeys obtained a free reward, and anti-saccade task in which the monkeys were required to make a saccade to the opposite direction of a peripheral visual cue). They first recorded LFP simultaneously from the subthalamic nucleus (STN) and the substantia nigra pars reticulata (SNr). They found that the phase angle of their LFPs changes between the tasks, suggesting that the weightings of the STN-SNr and STN-GPe-SNr pathways are changed dependent on the tasks. Second, they electrically stimulated the STN and examined the effect of the stimulation on eye movements during the tasks. They found that the effect of the STN stimulation is opposite between the tasks. This result suggests that the weighting of the STN-GPe-SNr pathway is dominant during the free viewing task and the weighting of the STN-SNr pathway is dominant during the anti-saccade task. These two experiments seem to suggest that the basal ganglia adaptively change the weighting of each pathway dependent on task demands. Their idea and findings are new and important to understand how the basal ganglia adaptively control motor action. But, I have some concerns about how to interpret their data.

We thank the Reviewer for their positive comments, and greatly appreciate their help in clarifying this manuscript. In addressing the Reviewer's comments, we have completed substantial revisions, and as requested, we have developed and included a spiking neural networks computational model of the basal ganglia. This is discussed in detail below.

Major concerns:

(1) They recorded LFP simultaneously from the STN and SNr. The phase angle of their LFPs was about 0 degree or slightly negative during the free viewing task. On the other hand, the phase angle was about 10 degree during the anti-saccade task. The authors explained that the increase in the phase angle occurs if the STN-SNr pathway becomes more dominant than the STN-GPe-SNr pathway. However, the increase was only 10 degree. As the authors predicted in the earlier part of the manuscript, the phase angle increase should be close to 180 degree if the STN-SNr pathway becomes more dominant. The authors discussed that the phase angle only slightly increased because of the effects of other pathways. But, I don't understand why other pathway can interrupt the full increase in the phase angle. A mathematical simulation using a simple neural network model of basal ganglia pathways could be useful for readers to understand why the increase in the phase angle was so small.

We thank the Reviewer for these comments, and the opportunity to clarify these findings. These necessitated the inclusion of a new computational spiking neural networks model, which has significantly strengthened the manuscript. For clarity, we have divided our response to the Reviewer into sections.

Prediction of a STN-SNr phase angle close to 180 degrees

We did not intend to predict an STN-SNr phase angle close to 180 degrees if the monosynaptic (hyperdirect) STN-SNr pathway was more active. As described in the Results (Line 88-97, 122-124), these predictions were based on the theoretical phase angle if only the STN-SNr pathway, or only the STN-GPe-SNr pathway, was active (in isolation). Specifically, we illustrated that a small positive phase angle might exist for the STN-SNr pathway (in isolation) because the STN activates the SNr but there is a signal transduction delay. In contrast, we suggested a negative phase angle close to 180 degrees (when measured peak-to-peak) might occur if the STN-GPe-SNr existed in isolation, because of the inversion of the signal through the GPe. In practice however, STN signals concurrently project to both the SNr and GPe (Castle et al., 2005 [J Comp Neurol]; Lanciego et al., 2012 [Cold Spring Harb Perspect Med]; Kita et al., 1983 [J Comp Neurol]; Van der Kooy and Hattori 1980 [J Comp Neurol]). In fact, in the Results (Line 122-124), we stated that we believed a 180 degree shift did not occur because of multiple simultaneous BG pathways. Both STN-SNr and STN-GPe-SNr pathways are active simultaneously (but with different weights, based on our data). Furthermore, activity in the SNr is strongly influenced by direct GABAergic projections from the caudate nucleus (i.e., “direct” BG pathway), and the STN is influenced by GABAergic projections from the GPe (i.e., “indirect” BG pathway). Therefore, the activity recorded in the STN and SNr do not solely represent subthalamo-nigral pathway signals. For this reason, we aimed to emphasize that the existence of any phase shift between task conditions indicates an intriguing shift in BG signaling, which was subsequently probed using STN/SNr/CD electrical stimulation results.

We have extensively revised the prediction of phase angle results to clarify this point (Lines 94-105). As suggested by the Reviewer, to further clarify this finding, STN-SNr phase angle was also simulated using a BG neural networks model which is described and referenced below.

Development of a basal ganglia neural networks model to test STN-SNr phase angle

As the Reviewer has indicated, the mechanism or rationale for a sub-180 degree change in STN-SNr phase angle between tasks may not be immediately clear to a reader. To clarify this, as the Reviewer requested we designed a BG model that artificially replicated STN, SNr, GPe, and CD neuronal activity, from which we could measure artificial STN-SNr coherence and phase angle. Importantly, to be conservative, BG activity in all relay (GPe, relay stage of STN) and output structures (SNr) were treated as a closed system, such that they did not receive any manual or pre-determined tonic activation input. We applied excitatory input to the STN and CD only (BG input nuclei), conceptually representing cortical input activity. Neural activity of other BG nuclei was determined by signal transmission from the STN and CD through the canonical direct/indirect/hyperdirect/subthalamo-pallido-nigral pathways. As now described in the Materials and Methods (Line 591-618), Figure 7, Abstract (Lines 13-15), Introduction (Lines 51-52), Discussion (Line 375-385), and Supplementary Figure 7, at estimates of STN and CD activity based on previous literature in free viewing and goal-directed saccade conditions, we observed a similar phase angle difference in artificial STN-SNr coherence as in LFP recordings. Specifically, artificial STN-SNr phase angle was negative during free viewing conditions, but positive during goal-directed conditions.

We suggested in the Discussion (Lines 352-369) that differences in BG input activation may mechanistically relate to the shift in BG pathway weighting downstream. In order to 1) reduce experimenter bias and ambiguity in task-specific BG input activity, and 2) test our hypothesis of altering BG input activity as a determinant of BG pathway weighting, we also examined artificial STN-SNr phase angle while independently varying both STN and CD activity from 0 to 100 spikes per second. As shown in Figure 7, artificial STN-SNr phase angle was consistently negative when STN and CD activity was low (as observed during free viewing conditions), while phase angle was consistently positive when STN and CD activity was high (as observed during goal-directed saccade conditions). This strongly suggested that 1) our physiologically-observed STN-SNr phase angle is reasonable, and 2) lends support to our hypothesis of BG input activity as part of the mechanism of action of altering BG pathway weightings.

(2) The authors showed the effect of electrical stimulation on saccade latency in “pro-saccade task” in Supplementary Materials (Figure S6). This is very important information. But the authors did not explain this data in the main manuscript. Furthermore, curiously, the effect of STN stimulation was the same (i.e., inhibitory) between the anti- and pro-saccade tasks. Although the pro-saccade task just required to execute a saccade to a visual target, anti-saccade task needs to a process that inhibits a saccade to a visual target. Therefore, it is reasonable that the STN-SNr pathway, which inhibits motor action, becomes more dominant during the anti-saccade task. Why did this pathway become more dominant even during the pro-saccade task? Author should show the results of the pro-saccade task in the main manuscript, and should explain why.

We thank the Reviewer for this comment. We are happy to include results of the pro-saccade task in the manuscript. As the Reviewer as requested, we have added pro-saccade analyses and results (Figure S3, S5, and S6) to allow for a complete comparison to free viewing and anti-saccade results.

BG stimulation effects during pro-saccades

We agree with the Reviewer’s rationale regarding why BG inhibitory activity is increased during anti-saccades. However, as described in the Introduction (Lines 55-62) and Discussion (Lines 330-339), and proposed for the SNr by Basso and Liu (2007) ([J Neurophysiol], “shunting inhibition”), we suggest that BG inhibitory activity is increased during a goal-directed task in order to inhibit sub-optimal movements (those directed away from the rewarding stimulus) in favor of robust voluntary movement activity. This is not limited to only the single erroneous visual stimulus presented during anti-saccades. In fact, the inhibitory effect of STN and SNr stimulation during pro-saccades supports this idea. While both STN and SNr stimulation increases pro-saccadic latency, there is a decreased magnitude with respect to anti-saccades. We therefore suspect that there is a continuum of task difficulties between free viewing, pro-saccade, and anti-saccade conditions, which is reflected in the results of BG stimulation. While pro-saccades are associated with fewer competing or distracting motor plans, there is nevertheless the necessity to execute a saccade toward a specific target to receive a reward, and we cannot discount the presence of other internal competing motor plans.

Location of new pro-saccade figures

We originally excluded a portion of pro-saccade data because the manuscript is already quite complex (e.g., 3 BG nuclei, 2 behavioral tasks, 2 experimental conditions). To address the Reviewer’s concerns we have included pro-saccade results to match anti-saccade results, as well as a BG spiking neural networks model (e.g., 3 BG nuclei, 3 behavioral tasks, 3 experimental conditions), but we would

like to emphasize our desire to maintain readability and a reasonable level of complexity in the manuscript. We are concerned with overwhelming a reader with results from too many experiments/conditions. Therefore, due to this and figure limits, we would prefer to leave the majority of pro-saccade analysis figures in Supplementary Figures, easily accessible to the interested reader. However, to address the Reviewer's comment, we have specifically discussed pro-saccades in the Results (Lines 235-244).

Other concerns:

(3) The authors showed the STN-SNr LFP coherence in Figure 3a. I want to see the “time course” of the coherence that is aligned by saccade onset. Heat map that includes all frequency bands would be nice.

We agree with the Reviewer that the STN-SNr LFP coherence time course may be interesting to investigate in the future. However, we specifically chose the LFP analysis time window based on 1) consistency with electrical stimulation analyses, and 2) to reduce or eliminate the confound of other signals such as visual responses, and reward responses or solenoid artifact from the juice system. We are unclear how this would strengthen the specific hypothesis proposed in this manuscript. At the least, this would prompt a duplicate analysis of all stimulation results at different temporal epochs, to continue to allow a comparison between these results. Given that we are at a premium of space, particularly after the very helpful inclusion of a BG spiking neural networks model, we would prefer to not include this analysis because we believe it will complicate the manuscript but not strengthen the results.

(4) The authors showed the phase angle before saccade onset. I want to see the phase angle around and after saccade onset, as well. The authors argued that the effects are tonic. If so, the same phenomenon should occur during other timings.

We thank the Reviewer for this comment. Unfortunately, this is not feasible because of the presence of non-physiological LFP artifacts from eye movement, and from reward-related signals immediately following the correct trial (both physiological and solenoid electrical noise). These contaminations prevent an accurate analysis of phase angle during the peri-saccadic period using LFP.

We would like to emphasize that while we hypothesized the BG effects might involve differences in tonic activation, we did not intend to suggest that acute saccade-related signals are not also transmitted through the BG (e.g., Hikosaka et al., 2000 [Physiol Rev]; Mink 1996 [Prog Neurobiol]; Nambu et al., 2002 [Neurosci Res]; Watanabe and Munoz, 2011 [Eur J Neurosci]). For this reason, if practical constraints did not prevent us from performing this comparison, we would nevertheless be concerned that the combination of tonic activity and acute saccade-related signals during the peri-saccadic period would be difficult to interpret.

Reviewer #3 (Remarks to the Author):

This manuscript describes a study of signalling pathways in the basal ganglia (STN and SNr) of monkeys, comparing between a free viewing saccade task and an instructed anti-saccade task. The hypothesis is that the relative strength of functionally opposing BG pathways – the

inhibitory STN-SNr pathway and the facilitatory STN-GPe-SNr pathway – are modulated based on task demands. Two approaches are used: First, the authors examine coherence and phase shift between pairs of simultaneous LFP recordings in STN and SNr, and find a phase shift difference of beta oscillations between free viewing and anti-saccade conditions. Second, the authors examined the effects of microstimulation in STN and SNr on the latency and hemifield bias of saccades. They found that while stimulation of SNr always inhibited saccades (contra or bilaterally, depending on site), stimulation of STN was task-dependent: facilitating contraversive saccades during the free viewing condition and inhibiting them during anti-saccade trials. Both approaches yield congruent conclusions, which support the hypothesis that the STN-SNr pathway dominates during the goal-driven saccades while the STN-GPe-SNr pathway dominates during free viewing.

Overall, this is a very solid study that provides important new information for making sense of basal ganglia circuitry. The study is well done and the paper is well organized, although there are some typos and missing information here and there, as well as some awkward writing. Experimental procedures are well planned and more generally, the study is technically sound. Statistical analyses are appropriately done and reported, though the authors' choice of different statistical tests is unclear and needs more justifications. I found the stimulation results much more compelling than the LFP analyses, but both are complementary.

We thank the Reviewer for these positive and helpful comments. In order to expand on the LFP analyses, and in response to a comment from Reviewer 2, we have now paired these with a BG spiking neural networks model (CD, STN, GPe, SNr). This specifically models STN-SNr phase angle, while varying BG input activity at the level of the STN and CD.

A general critique is that this study does not really address “task-dependence” in the broad sense, but instead addresses the more specific issue of “inhibition” of automatic responses (clearly an important strategy to prevent premature responses in anti-saccade tasks). I would suggest the authors frame their results in terms of inhibition instead of in a way that appears to pertain to task-dependence in general, especially since the role of the STN in control of inhibition has already been well-established. The major difference between the tasks is that the anti-saccade task demands the withholding of immediate responses, engaging deliberate inhibition. So the difference is not a general “task-dependence”, but a specific difference in inhibitory control that the monkey learned through training. In fact, I’d recommend making this clear throughout by describing those trials as “anti-saccade” instead of “goal-directed”, in all figures and text. Furthermore, for this reason it would be useful to look at pro-saccades. Since they are interleaved among anti-saccades they might also be more inhibited than pure free viewing. Indeed, STN-SNr coherence in pro-saccades (Fig S4) looks pretty similar to coherence in anti-saccades (Fig S3). But what were the effects of SNr and STN stimulation during the pro-saccade trials? How did they differ from the anti-saccade condition, and do they make sense in the context of the scheme proposed in Figure 7? I agree with the authors' motivation for focusing on anti-saccades because of the absence of a visual stimulus at the saccade target site, but looking at the pro-saccade data might nevertheless be useful. After the fixation color appears, and indicates a pro-saccade trial,

the inhibitory control might be relaxed a bit (like the free-viewing trials) but nevertheless the saccade is still goal-directed (like anti-saccade trials).

We thank the Reviewer for the opportunity to correct and clarify this. We have subdivided our responses below into, first, a discussion of pro-saccade results, and second, framing the results in the context of task-dependence and/or an inhibition of automatic responses.

Inclusion of pro-saccade results

We agree that including pro-saccade analyses would strengthen this manuscript, so we have added these as indicated by the Reviewer. Pro-saccade results can now be directly compared to free viewing and anti-saccade results (Supplementary Figures 3, 5, 6; Results, Lines 235-244). As described in the revised manuscript, and predicted by the Reviewer, pro-saccade results were qualitatively similar to anti-saccade results. While anti-saccades require the withholding of an automatic visually-guided saccade and pro-saccades do not, importantly both require saccades to be directed toward a specific target to receive a reward. This is in stark contrast to free viewing behavior, which was voluntarily executed, but not associated with any explicit reward. This suggests that the shift in BG pathway weighting may be related to unconstrained versus goal-directed saccade behaviors.

Task-dependence and/or inhibition of automatic responses

As the Reviewer notes, we chose to frame our results with respect to task-dependence and not an inhibition of automatic responses. This was in fact because of pro-saccade results, which we now include in the manuscript. Task characteristics and results are summarized below:

Task Condition	Automatic or Voluntary	Unconstrained or Goal-Directed	Stimulation Effect
Free viewing	Voluntary	Unconstrained	Facilitation
Anti-saccade	Voluntary Errors are automatic saccades toward the visual target	Goal-Directed	Inhibition
Pro-saccade	Combination of voluntary and automatic Visually-guided saccades are directed toward the rewarded goal	Goal-Directed	Inhibition

As summarized here, pro-saccade and anti-saccade stimulation results were qualitatively similar, even though pro-saccades are a combination of voluntary and automatic movements and anti-saccades are

purely voluntary. We agree with the mechanism predicted by the Reviewer during pro-saccades (i.e., inhibitory control is reduced after the pro-saccade fixation color appears, but is nevertheless still present because pro-saccades are also goal-directed). This is supported by pro-saccade results (Fig. S3, S5, S6), which were qualitatively similar to anti-saccade results but with a decreased absolute magnitude of effect. However, we are concerned that the interpretation of inhibition of automatic responses might not fully reflect our results. Unlike both free-viewing and anti-saccade conditions, pro-saccades are a combination of voluntary and automatic movements (according to whether the visual response is directly transformed into a saccade initiation cue; e.g., Munoz and Everling, 2004 [Nat Rev Neurosci]). This suggests that BG stimulation effects are not limited to inhibiting automatic responses. Otherwise, we might expect pro-saccade and free-viewing/anti-saccade responses to be strikingly different, which was not the case. We therefore find it most plausible that the BG modulated inhibition according to unconstrained or goal-directed behaviors, as described in the Introduction (Lines 52-62) and Discussion (Lines 330-341), in order to reduce sub-optimal movements in favor of goal-directed movements with robust voluntary responses (as proposed for the SNr by Basso and Liu (2007) [J Neurophysiol], “shunting inhibition”). This may include, but is not limited to, the inhibition of automatic movements. Furthermore, pro-saccades and anti-saccades are frequently described by the terminology “automatic” and “voluntary,” respectively. We are concerned that this ambiguity of nomenclature could introduce confusion to the reader.

For these reasons, we prefer to take the conservative stance that BG activation is task dependent (defined as a difference in neural activity and/or stimulation effect according to task condition) between unconstrained and goal-directed behaviors. This is a subtle difference from the Reviewer’s suggestion of inhibition of automatic responses, which we nevertheless believe is relevant. Importantly, a task-dependent switch according to unconstrained or goal-directed behaviors does not preclude the specific inhibition of automatic responses.

It might be useful to look at LFP coherence and phase during the fixation period, prior to cue appearance.

We thank the Reviewer for this comment. However, there are practical limitations preventing this analysis. First, a fixation point exists during pro- and anti-saccades, but not during free viewing. This difference in visual signals directly confounds a comparison between free viewing and pro-/anti-saccade tasks during the fixation period. Second, we cannot compare these LFP results to stimulation results (because stimulation was applied in the pre-saccadic period, not during fixation). Therefore, results from this analysis would be difficult to interpret, and we are concerned would not strengthen the conclusions of the manuscript.

The authors predict a 180 deg STN-SNr phase difference between the free-viewing and the anti-saccade condition. If I understand it correctly, this is because of an expected change in sign when the STN-GPe-SNr pathway is stronger than the STN-SNr pathway. But that’s not what is seen. The phase difference is much smaller and rather weak. For most sites (except two: Fig 2b), the phase shifts are distributed around 0. How do the authors explain this discrepancy? To me, it seems that phase shifts could be produced by changes in sign, or by changes in transmission delays, or both – making interpretation challenging. My understanding is that beta oscillations are a product of the entire BG loop through cortex and thalamus, so I would have predicted only slight phase shifts depending on whether the signal gets through the STN-SNr or the (presumably

slightly slower) STN-GPe-SNr pathway. A 180 deg shift would be difficult to imagine (it would correspond to a delay of ~25ms). Perhaps I don't understand the logic behind the authors' interpretation of the phase shifts, in which case I'd appreciate more clarification.

The authors discuss their results in terms of the "comparative weighting of BG pathways" (e.g. line 5), but that phrase makes it sound like a hypothesis about changes in synaptic connectivity. As far as I can tell, the authors are really suggesting that during anti-saccade trials, there is more inhibition of GPe by the caudate, reducing the possibility for STN to inhibit the SNr and instead allowing its facilitatory pathway to dominate. Is that right or are the authors trying to argue for something more?

We thank the Reviewer for the opportunity to clarify this. We hypothesized that if there are task-dependent alterations in the tonic weighting of BG pathways (such as between the hyperdirect and subthalamo-pallidal-nigral pathways), there would be a change in STN-SNr phase angle because of a different proportion of signals from pathways with 1) different signal transduction times, and 2) potentially a 180° signal phase shift (sign inversion) of STN efferent signals which pass through the GPe. A dominance of the hyperdirect pathway may be associated with a positive STN-SNr phase difference (based on glutamatergic subthalamo-nigral projections and a short signal transduction time), while a dominance of the subthalamo-pallidal-nigral pathway may be associated with a negative STN-SNr phase difference (when measured peak-to-peak; based on a 180° signal phase shift, less the increased subthalamo-pallidal-nigral transduction time; see new Figure 1a). If STN-SNr and STN-GPe-SNr pathways existed in isolation, and signals projected 100% through one or the other pathway, then we might expect a small positive STN-SNr phase difference for the monosynaptic STN-SNr pathway (e.g., STN-SNr signal transduction delay), and a close to 180° signal phase shift for the STN-GPe-SNr pathway because of the sign inversion through the GPe (e.g., STN-GPe-SNr signal transduction delay combined with a sign inversion). However, it is important to note that 1) even in an ideal theoretical situation, STN-GPe-SNr signals would not be associated with a perfect 180° phase shift due to the combination with a signal delay through the GPe, and 2) activity in the STN and SNr reflects multiple pathways in addition to the hyperdirect and subthalamo-pallido-nigral signals. In particular, STN activity includes both hyperdirect and indirect pathway signals, and SNr activity reflects signals from all major canonical pathways. For this reason, we conclude that these LFP results are a strong indication that there is a task-dependent difference in BG signaling, but we only hypothesize this is may be specifically due to STN-SNr versus STN-GPe-SNr pathways. We further test this using electrical stimulation of the STN, SNr, and CD. Furthermore, we have now supplemented LFP phase angle results with a BG spiking neural networks model. This was supportive of our suggestion in the Discussion that biases toward facilitatory or inhibitory BG pathways are mediated by decreased or increased excitatory input to the BG (STN and CD), respectively. We have re-written the description of LFP phase analysis in the Results to clarify this point (Lines 91-105, 118-124)

We find it unlikely that changes in synaptic connectivity would occur fast enough to underlie the task-dependent results presented here. Based on the LFP recording, electrical stimulation, and spiking neural networks modeling results, we can only conclude that BG pathways have different relative influences on SNr activity according to task condition, and we can causally link this to saccade behavior using electrical stimulation. As the Reviewer has suggested, we have clarified this point in the Abstract by no longer using the term "comparative weighting" (we have replaced this with "relative strength"; Line 5).

SPECIFIC COMMENTS

Isoda & Hikosaka (2008) already made a good case that the STN is involved in task switching. That paper is cited in the methods, but it seems relevant for the introduction and discussion as well.

We have now referenced this paper in the Introduction (Lines 25-26) and Discussion (Lines 339-341) as the Reviewer has indicated.

The numbers of sites where a significant effect of SNr stimulation has been found are confusing. For instance, in line 138 it is stated that out of 68 sites, 47 produced only contraversive inhibitions. But in line 162, it is stated that stimulation significantly affected contraversive saccade initiation in 18 sites. This should be clarified.

We apologize for this confusion. $n=47/68$ reflected all stimulation sites that did not exhibit a bilateral inhibition, not the number of sites that exhibited contraversive inhibition. We have clarified this in the Results (Lines 144-146).

It's not clear how the authors determine online, for the microstimulation, the 200ms pre-saccadic period in the free-viewing task.

We did not determine a 200msec period online in the free-viewing task. Electrical stimulation during free viewing was applied in the following way (described in the Methods, Lines 461-474). During free viewing, monkeys made saccadic eye movements across the grey screen whenever they pleased. "Trials" were defined online, but were unknown to the monkey. On each "trial," we set an invisible computer-controlled window ($\pm 10^\circ$) on the center of the screen and waited for up to 30 s for the monkey to happen to direct gaze into the window. We analyzed the first saccades initiated at least 300 ms after the eyes entered the window. We excluded trials with saccades initiated between 0ms to 300ms after the eyes entered the window, to remove step saccades that may have passed only briefly through the window. On half of the trials, electrical stimulation was initiated 300 ms after eyes entered the window and lasted until the end of the trial. We used a velocity and displacement-based detector to detect saccades online, which prevented and/or cancelled electrical stimulation after saccade initiation. Trials were ended after a saccade was initiated or the eyes left the window (saccade trials), or 800 ms after the eyes entered the window (no saccade trials). As described in the Methods, mean stimulation duration was 253/238ms (SD=33/27ms) for monkey E STN/SNr, respectively, and 249/239ms (SD=21/24) for monkey O STN/SNr, respectively.

To clarify this, we have revised Lines 135-137 in the Results section, and created a figure that illustrates the time sequence of task events, eye movements, LFP recording period, and stimulation period (Supplementary Figure 1).

How did the authors choose between stimulation intensities (between the 10-40 μ A ranges)? It is not clear if the authors excluded free saccades that did not match the spatial characteristics of the ones studied in the anti-saccade task. How large was the tolerance window in the anti-saccade task, and was it comparable to the saccades examined in free-viewing?

At each site, we tested up to 40 μ A stimulation. As described in the Methods (Lines 546-561), at higher currents (30-40 μ A) some STN or SNr stimulation sites resulted in tightly clustered saccade vector endpoints in the free viewing task, and the monkey was only able to produce saccades toward one hemifield in the goal-directed saccade task regardless of task instruction. At these sites, the current amplitude was lowered below 40 μ A in order to record enough of both ipsiversive and contraversive correct goal-directed trials for the analyses of reaction times.

In the free viewing results, we presented stimulation results of all saccades from all stimulation sites, to be as transparent as possible with our data. However, to address the Reviewer's comment, we have also re-analyzed free-viewing STN and SNr stimulation effects while stringently removing all saccades with endpoints directed outside a $\pm 3^\circ$ square bounding window centered on 12° eccentricity. This matched anti-saccade stimulus locations and tolerance window size (Methods, Line 493-495). We found no significant difference in stimulation latency effects for these filtered free-viewing saccades compared to the full population. We have indicated this in the Methods (Lines 479-484).

It is not clear how the authors choose their statistical tests to evaluate the effect of stimulation on saccade parameters. A t-test is used to assess the effect of microstimulation on saccade endpoint distributions, but a KS-test for the effect of stimulation on saccade frequency. More problematically, the KS-test is sometimes used to test the effect of stimulation on saccade latencies (line 173-174), but most of the time the authors use t-tests (for example in line 158). Why??

A Kolmogorov-Smirnov test was used to assess the effect of stimulation on saccade frequency only. The Kolmogorov-Smirnov test is a nonparametric test of the null hypothesis that two sample datasets are derived from the same population. This was most appropriate to examine differences in saccade frequency, because free-viewing saccade frequency distributions were uniformly distributed, not normally distributed (see Discussion, Lines 341-344, and Fig S4).

Line 173-174 referenced by the Reviewer from the original manuscript described a comparison of saccade frequencies (across different latencies), not a comparison of saccade latencies. We have revised this sentence to clarify that the statistical test pertains to saccade frequency, and not latency (Lines 180-184).

Talking about "saccade latency" in the free-viewing condition is rather strange, since there is no known trigger with which to compare saccade onset. The authors instead use the start of the last central fixation – but then the quantity being measure is actually fixation duration. I understand the desire to compare these to the latencies in the anti-saccade task, but it would be important to acknowledge the differences between measures and potential confounds those differences could introduce.

We thank the Reviewer for this comment. As described in the Results (Lines 197-200), we define free viewing saccade latency as the time delay from electrical stimulation onset to saccade initiation. Saccade latency, as defined in this manuscript, is therefore not synonymous with fixation duration. We apologize for this confusion, and have clarified this definition in the Results (Lines 198-200), Methods (Lines 577-580), and have illustrated this in Supplementary Figure 1.

Lines 22-23: “direct” and “indirect” should be reversed.

We indicated in old Lines 22-23 that direct and indirect pathways inhibit or activate BG output nuclei (such as the SNr), respectively. This is correct, however we have clarified in Lines 22-24 that we were referring to BG output nuclei, and not downstream structures such as the SC that are modulated by BG output signals.

Line 95: typo: glutamatergic instead of glutamatertic.

We thank the Reviewer, we have corrected this typo.

Line 105: Why is it $p < 0.05$ for a 99% confidence interval?

We thank the Reviewer, we have corrected this mistake in the text.

The scale of the phase variable in Figure 3a could be more informative. In particular, it's important to know where the value of 0 lies (it doesn't even appear to correspond to a tick-mark).

We have added arrows to indicate the value of 0 on Figure 3a. We agree that this improves the readability of this Figure.

Line 180: I would replace the title of the section with: Effect of STN stimulation was task dependent

We have replaced the title of the section as indicated by the Reviewer.

Line 256-257: Should be “subthalamo-pallido-nigral” pathway

We thank the Reviewer, we have corrected this typo.

Line 312: Don't you mean “STN stimulation”?

Caudate stimulation is correct in (former) Line 312. This paragraph referenced by the Reviewer was describing the caudate nucleus stimulation results, which were previously reported in the same monkeys.

Lines 331-332: “while hyperdirect and STN-GPe-SNr pathways may mediate global effects.” This part of the sentence is not explicit enough. The authors should at least briefly describe some of these “global” effects.

Global effects (referenced by Aron 2011 [Biol Psychiatry]) referred to a general inhibition or disinhibition across the retinotopic map, rather than directed toward more specific spatial locations. We have clarified this statement in the Discussion (Line 348-349).

When comparing the results of the present experiment with those on the caudate published previously (lines 311-313 of the discussion), did the authors notice a difference of effect strength (quantitatively) between the stimulation of the caudate and the STN in the free-viewing task? Both induce a contraversive bias but I was wondering if the weighting between the classic direct pathway (CD-SNr) and the STN-GPe-SNr pathway is different.

STN and caudate nucleus stimulation results can be directly compared in Figure 5. The magnitude of STN stimulation effect was significantly larger on ipsiversive pro- and anti-saccades compared to caudate nucleus stimulation (paired *t*-test, $p < 0.01$). There was no significant difference between STN and caudate stimulation effects during any other condition.

Are the pairs shown in Figures S2-4 the same in each corresponding panel? If so, then it could be useful to overlay the three conditions (each with different colors) in the same figure, and discuss the differences.

We thank the Reviewer for this comment. The Reviewer is correct, these pairs were the same in each corresponding panel in (former) Figures S2-4. As the Reviewer has indicated, we have overlaid these conditions (Supplementary Figure 3), which we agree improves the readability of these data.

The population effects described in Figure 2b are observable in the individual sites in Fig S3. There was strong coherence between STN and SNr signals encompassing the beta frequency band during both free viewing and pro-/anti-saccades; from approximately 5-29 Hz during pro-/anti-saccades but from 5-48 Hz during free viewing saccades. STN-SNr coherence was also significantly higher in the free viewing task than pro- and anti-saccades from 31-58 Hz (Wilcoxon Rank-Sum Test, $p < 0.05$; Fig. 3a). There was no significant difference in STN-SNr coherence between the populations of pro- and anti-saccade conditions.

REVIEWERS' COMMENTS:

Reviewer #1 (Remarks to the Author):

Overall, the manuscript is improved and I have only one comment (2).

- 1) The new diagrams in Fig 1 and 7 are now much improved.
- 2) I suggest to move Fig S1 to Fig 1c. The time sequence of the task events and stimulation is crucial. For example, as I pointed out previously, the effect of stimulation may be different depending on the stimulation timing relative to the initiation of saccades. It is also critical to understand the temporal relation between electrical stimulation and 'phase'.

Reviewer #2 (Remarks to the Author):

The authors have responded sincerely to my concern and have improved the manuscript. This is a very nice study and I am looking forward to seeing it in print.

Reviewer #3 (Remarks to the Author):

The authors have addressed all of my concerns very well. In particular, they have convincingly argued that the modulation they observe is indeed an example "task-dependence", and not merely an "inhibition of automatic responses" as I had proposed. The key is the inclusion of data from the pro-saccade trials, which show similar phenomena as data from the anti-saccade trials, albeit with some delays and reduction of effects. This is just what would be expected if the difference between pro vs. anti is a difference in withholding, but the difference between pro-or-anti vs. free viewing is a more profound dependence on being goal-directed or not, as proposed by the authors. The only thing I'd add (and this is completely up to the authors' discretion) is a brief discussion of the role of the STN in changes of the speed-accuracy tradeoff, as proposed by Michael Frank, Rafal Bogacz and others. From that perspective, it is plausible to suppose that in free viewing the brain is in a more speedy regime than during the instructed pro/anti-saccade task, and this might be at least partially responsible for the differences seen here between the AS and FV tasks.

My other general comment was on the interpretation of the STN-SNr phase angle difference, which is smaller than what I would have expected if there was a significant modulation of the pathways, one of which has a sign inversion. The authors make a good argument that this would not be expected even in the ideal theoretic situation, and provide a model to demonstrate this point. That is indeed helpful.

However, while the model is useful to demonstrate the interpretation of the phase angle results, it does not seem as a very thorough "model of the STN, CD, GPe, and SNr" as described in lines 51-52 and elsewhere. It doesn't actually simulate any functional aspect of the tasks, and is not really a model of spiking behavior (e.g. integrate-and-fire). Instead, it seems to be a set of very basic binary McCulloch-Pitts neurons doing Boolean arithmetic, and its only purpose is to show what happens as you vary the strength of a sign-inverted pathway in something whose dynamics are surely very different than the real BG. It's strange to see that only McCulloch & Pitts are cited in the context of a BG model (line 377) – Their paper didn't say anything the BG, and since then there have been many other much more realistic models (e.g. the work of Michael Frank, Rafal Bogacz, Joshua Brown, etc). Note that I'm not suggesting that the authors add a more sophisticated model, but just to more clearly present what this model is and is not meant to simulate. Personally, I think that including the

model helps to answer one question, but introduces so many others that I wonder if it's ultimately useful in the end. Nevertheless, I understand that the model was included to respond to the comments of reviewer #2, so I'll leave it up to that reviewer and the authors to decide whether the model earns its place in the paper.

In short, I think this is a very good paper and I have only minor comments and optional suggestions.

We thank the reviewers and editors for all of their valuable comments on our original manuscript entitled “Evidence for a task-dependent switch in subthalamo-nigral basal ganglia signaling” throughout the submission to *Nature Communications*. We believe the manuscript has been substantially improved by taking into account these suggestions, and we hope that it is now acceptable for publication in *Nature Communications*.

In the following sections, red sentences indicate the comments of the reviewers and black sentences indicate our point-by-point replies to these comments. Revisions made in the main text are recorded by tracked changes. Line numbers referenced below refer to the tracked changes document.

REVIEWERS' COMMENTS:

Reviewer #1 (Remarks to the Author):

Overall, the manuscript is improved and I have only one comment (2).

- 1) The new diagrams in Fig 1 and 7 are now much improved.
- 2) I suggest to move Fig S1 to Fig 1c. The time sequence of the task events and stimulation is crucial. For example, as I pointed out previously, the effect of stimulation may be different depending on the stimulation timing relative to the initiation of saccades. It is also critical to understand the temporal relation between electrical stimulation and ‘phase’.

We thank the Reviewer for their positive comments here, and throughout the submission process. As indicated by the Reviewer, Fig S1 is now presented as Fig 1c.

Reviewer #2 (Remarks to the Author):

The authors have responded sincerely to my concern and have improved the manuscript. This is a very nice study and I am looking forward to seeing it in print.

We thank the Reviewer for their positive comments.

Reviewer #3 (Remarks to the Author):

The authors have addressed all of my concerns very well. In particular, they have convincingly argued that the modulation they observe is indeed an example “task-dependence”, and not merely an “inhibition of automatic responses” as I had proposed. The key is the inclusion of data from the pro-saccade trials, which show similar phenomena as data from the anti-saccade trials, albeit with some delays and reduction of effects. This is just what would be expected if the difference between pro vs. anti is a difference in withholding, but the difference between pro-or-anti vs. free viewing is a more profound dependence on being goal-directed or not, as proposed by the authors. The only thing I’d add (and this is completely up to the authors’ discretion) is a brief discussion of the role of the STN in changes of the speed-accuracy tradeoff, as proposed by Michael Frank, Rafal Bogacz and others. From that perspective, it is plausible to suppose that in free viewing the brain is in a more speedy regime than during the instructed pro/anti-

saccade task, and this might be at least partially responsible for the differences seen here between the AS and FV tasks.

We thank the Reviewer for their positive and helpful comments. As suggested by the Reviewer, we have added a brief discussion of the involvement of the STN and BG to the speed-accuracy tradeoff (Discussion, Lines 399-410).

My other general comment was on the interpretation of the STN-SNr phase angle difference, which is smaller than what I would have expected if there was a significant modulation of the pathways, one of which has a sign inversion. The authors make a good argument that this would not be expected even in the ideal theoretic situation, and provide a model to demonstrate this point. That is indeed helpful.

However, while the model is useful to demonstrate the interpretation of the phase angle results, it does not seem as a very thorough “model of the STN, CD, GPe, and SNr” as described in lines 51-52 and elsewhere. It doesn’t actually simulate any functional aspect of the tasks, and is not really a model of spiking behavior (e.g. integrate-and-fire). Instead, it seems to be a set of very basic binary McCulloch-Pitts neurons doing Boolean arithmetic, and its only purpose is to show what happens as you vary the strength of a sign-inverted pathway in something whose dynamics are surely very different than the real BG. It’s strange to see that only McCulloch & Pitts are cited in the context of a BG model (line 377) – Their paper didn’t say anything the BG, and since then there have been many other much more realistic models (e.g. the work of Michael Frank, Rafal Bogacz, Joshua Brown, etc). Note that I’m not suggesting that the authors add a more sophisticated model, but just to more clearly present what this model is and is not meant to simulate. Personally, I think that including the model helps to answer one question, but introduces so many others that I wonder if it’s ultimately useful in the end. Nevertheless, I understand that the model was included to respond to the comments of reviewer #2, so I’ll leave it up to that reviewer and the authors to decide whether the model earns its place in the paper.

In short, I think this is a very good paper and I have only minor comments and optional suggestions.

We agree with the Reviewer regarding a specific scope for the model presented here. We believe the model does strengthen the manuscript. However, as described in the Discussion (Lines 411-421) and Methods (639-664), this model was designed only to test the plausibility of subthalamo-nigral phase angle changes, due to tonic activation signals. To address the Reviewer’s concern, we have carefully reviewed and edited the manuscript to clarify the design and aim of the model (Introduction, Lines 51-52; Discussion, Lines 411-412; Methods, Lines 635-640, 661-664).

We chose a simple McCulloch-Pitts neuron as an elementary unit to construct a BG model because it is easily understandable, it limits complexity that may detract attention from what is otherwise a neurophysiological paper, and it allows an easy interpretation of its results. Similar to the Reviewer, our concern was that the multitude of parameter settings in more complex models might add tangential questions, which we would be unable to address within the scope and word limits of this manuscript.